# Prompt-guided Disentangled Representation for Action Recognition

**Tianci Wu, Guangming Zhu,**\* **Jiang Lu, Siyuan Wang, Ning Wang,**
**Nuoye Xiong, Zhang Liang**
School of Computer Science and Technology, Xidian University
{tianciwu, lujiang, siyuanwang, ningwang, nyx}@stu.xidian.edu.cn
{gmzhu, liangzhang}@xidian.edu.cn

## Abstract

Action recognition is a fundamental task in video understanding. Existing methods typically extract unified features to process all actions in one video, which makes it challenging to model the interactions between different objects in multi-action scenarios. To alleviate this issue, we explore disentangling any specified actions from complex scenes as an effective solution. In this paper, we propose Prompt-guided Disentangled Representation for Action Recognition (ProDA), a novel framework that disentangles any specified actions from a multi-action scene. ProDA leverages Spatio-temporal Scene Graphs (SSGs) and introduces Dynamic Prompt Module (DPM) to guide a Graph Parsing Neural Network (GPNN) in generating action-specific representations. Furthermore, we design a video-adapted GPNN that aggregates information using dynamic weights. Extensive experiments on two complex video action datasets, Charades and SportsHHI, demonstrate the effectiveness of our approach against state-of-the-art methods. Our code can be found in `https://github.com/iamsnaping/ProDA.git`.

## 1 Introduction

Action recognition is one of the fundamental tasks in the field of video understanding [16], and it remains an active topic in the vision research community. Current mainstream methods can be categorized into two types based on their data processing approach. The first type is based on complete video data. These methods typically input the untrimmed video into 2D Convolution Neural Network (CNN), 3D CNN, or Vision Transformer (ViT) [7], using the network to extract features and then performing classification using the extracted features [10, 42, 4, 45, 27, 30, 27]. The second type is object-centric, where object detectors first crop objects from the video and then input them into ViT or Graph Neural Network (GNN) [53, 33, 57, 48, 54, 55, 48, 53]. Some methods [59, 20, 47, 35] also construct Spatio-temporal Scene Graph (SSG) from these objects and then feed them into the network. However, these methods are essentially information extraction processes that handle all actions in the video at once, making it difficult to model the interactions between objects in complex scenes [48, 55]. In this paper, we aim to analyze actions with a disentangled approach. For instance, in a video with multiple actions, we want the network to focus on analyzing some specified action(s). Moreover, we construct SSGs for this video and aim to disentangle the subgraph, which consist of the specified action-relevant nodes, enabling better understanding of the actions (as shown in Figure 1). Existing Disentangled Representaion Learning (DRL) methods typically approach disentanglement from different perspectives. Some methods focus on attribute disentanglement, aiming to separate content from background [22, 52, 17]. Others adopt an information perspective by disentangling representations into structural invariants and local variations (e.g., shared information

---

\*Corresponding Author

39th Conference on Neural Information Processing Systems (NeurIPS 2025).

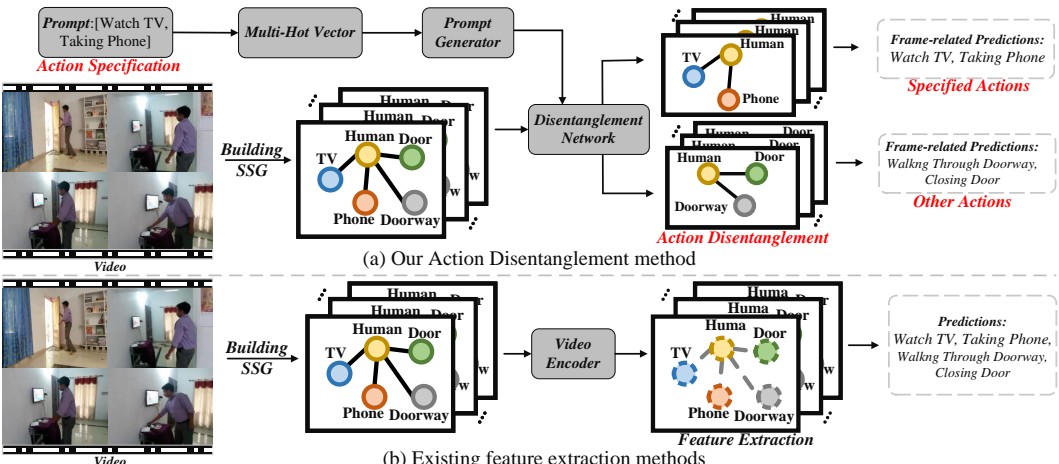

(a) Our Action Disentanglement method

(b) Existing feature extraction methods

Figure 1: Comparison between (a) our proposed Prompt-guided Representation Disentanglement method and (b) existing methods using SSG. Existing methods treat action modeling as a feature extraction process, handling all actions in one video. In contrast, our method disentangles the action representation into two parts: one for the specified action and the other for the remaining actions, enabling more focused analysis and better action understanding.

across nodes and private, node-specific features) [6, 58, 12, 34]. From a structural perspective, some approaches disentangle graphs into a fixed number components(e.g., latent representation, subgraphs) [26, 25, 24, 6, 58, 12]. A task perspective has also been explored, where complex tasks are divided into multiple subtasks to facilitate more structured representation learning [29]. In action recognition, **we argue that disentanglement should be approached from the action perspective**, enabling the dynamic separation of semantic factors related to *any specified actions*. Specifically, we expect the network to analyze any actions we specify and to disentangle the corresponding subgraph from the whole SSG, enabling more targeted and interpretable action understanding.

To this end, in order to enable the model to disentangle representations conditioned on any specified action, we draw inspiration from recent advances in prompt tuning [31, 8, 39, 21, 56, 2] and propose Prompt-guided Disentangled Representation for Action Recognition (ProDA). As depict in Figure 1 (a), **ProDA disentangles the SSG into two complementary parts: one corresponding to the specified actions, and the other to the remaining actions.** To achieve this, we construct an action-aware disentanglement network that can disentangle any specified actions from SSG based on prompt guidance. Specifically, we introduce the Dynamic Prompt Module (DPM), which generates action-aware prompts by combining the multi-hot vector of specified actions with the SSG features. These prompts guide the Graph Parsing Neural Network (GPNN) to perform action-aware disentanglement. Furthermore, we propose a Video Graph Parsing Neural Network (VGPNN), a GPNN variant tailored for SSG, to perform action disentanglement. To ensure temporal consistency and preserve the differences across SSGs from different videos, we introduce a Video Graph Normalization (VGNorm) module, which is applied after each VGPNN layer. Finally, we introduce an Action Disentanglement Loss (AD Loss) to supervise the separated representations. To ensure that the disentangled features retain essential information, we incorporate a reconstruction term, promoting true representation disentanglement rather than mere feature extraction.

In summary, the major contributions of this work are as follows: 1) We present a novel framework ProDA, which disentangles any specified action from multi-action videos, enhancing the interpretability of action understanding; 2) Our method achieves State-of-The-Art performance on multi-label action recognition benchmark. 3) Without requiring action localization supervision, our method demonstrates strong action localization capabilities, further validating its effectiveness and interpretability in understanding complex actions.

## 2 Related works

Existing action recognition methods can be broadly categorized into three types. First type is CNN-based methods. To compensate for the lack of temporal modeling in 2D CNNs, two-stream networks introduce an additional temporal stream (e.g., optical flow)[10, 9, 41, 44, 49]. By jointly modeling

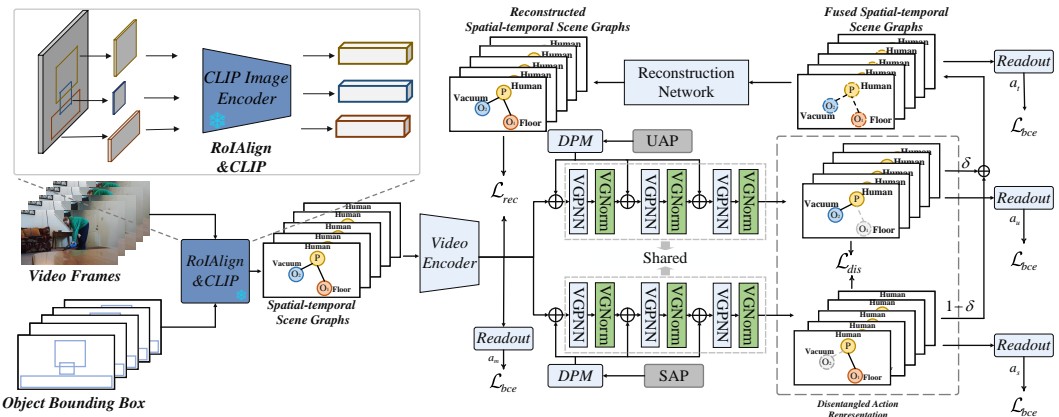

Figure 2: An overview of our ProDA. We first convert the input video into a sequence of scene graphs, which are then processed by a Video Encoder. Next, we generate two complementary prompts from the Action Set (SAP and UAP) using DPM to guide the VGPNN in disentangling the actions into two complementary SSGs, each containing distinct parts of the action. These disentangled SSGs, along with their fused version, are fed into corresponding readout blocks to obtain the global representations of partial and complete actions, respectively.

spatial and temporal features, 3D CNNs are able to better capture spatio-temporal information and thus yield improved performance [42, 4, 51]. The second type is ViT-based methods. Most methods either treat entire frames as input tokens or divide each frame into patches before feeding them into the model [13, 45, 27, 30, 27]. These methods treat the video as a whole, which makes it difficult to model interactions among objects, and often introduces substantial noise. Our work alleviate this by disentangling actions from the SSG, enabling more precise modeling of object interactions. The third type is Objected-centric methods. Object-centric methods construct more compact and efficient video representations by focusing on regions of interest (e.g., objects in the video) [53], and enabling generalization of unseen interaction relationships through structured representations [53, 33, 57, 48, 59]. Nevertheless, these methods remain limited in their ability to effectively model interactions among objects. The incorporation of SSG [59, 20] effectively addresses the difficulty in modeling object interactions. In scene graphs, videos are represented as spatio-temporal graph structures, with objects serving as nodes (e.g., human, cup) and their interactions or spatial-relationship (e.g., holding, above) modeled as edges. [35] processes the scene graph of each frame using a Graph Neural Network (GNN). On the other hand, [47] flattens the scene graphs and feeds them into a ViT [7], leveraging the long-range modeling capability of self-attention [43] to effectively capture interaction transitions. These SSG-based methods still adopt feature extraction approaches that process all actions in a video, which makes it challenging to model interactions between different objects in multi-action scenarios. In contrast, our method disentangles actions from complex scenes, enabling better understanding of the actions

## 3 Methodology

Our method aims to disentangle any specified actions from multi-action scenarios. This disentanglement provides more concise feature representation for analyzing the specified actions, leading to improved action understanding. The overall architecture of our method is illustrated in Figure 2, we first convert the input video into a SSG. This graph is then processed by a video encoder (specifically, Dual-AI [15]) to model temporal and spatial interactions among nodes within the SSG. Next, guided by the DPM, the video features are fed into VGPNN for representation disentanglement. Finally, to further enhance the disentanglement, in addition to standard classification supervision, we introduce both a reconstruction loss and a disentanglement loss as constraints. We generate the SSG [38] using existing methods and adopt a off-the-shelf video encoder [15]; the following focuses on the remaining components of the network architecture illustrated in Figure 2.

### 3.1 Formulation

In SSG, each frame of a video is processed into a graph, where nodes represent the objects present in that frame, and edges represent the semantic relationships between these objects (e.g., spatial or

interaction relations). Formally, let $\mathcal{G} = (\mathcal{V}, \mathcal{E}, \mathcal{R})$ denote the complete SSG constructed from the video, where $\mathcal{V}$ is the set of all object nodes across all frames, $\mathcal{E}$ is the set of all directed edges, and $\mathcal{R}$ is the set of all relation types. Each node $v \in \mathcal{V}$ is uniquely indexed from $\{1, \ldots, |\mathcal{V}|\}$. Each edge $e = (v, w, r) \in \mathcal{V} \times \mathcal{V}$ represents a directed semantic relation of type $r \in \mathcal{R}$ from node $v$ to node $w$. Let $g_i = (\mathcal{V}_i, \mathcal{E}_i, \mathcal{R}_i)$ denote the scene graph of the $i$-th frame, where $\mathcal{V}_i \subseteq \mathcal{V}$, $\mathcal{E}_i \subseteq \mathcal{E}$ and $\mathcal{R}_i \subseteq \mathcal{R}$ denote the frame-specific object nodes, their edges and relations. We denote $v_i^j$ as the $j$-th object node in frame $i$, and $e_i^{j,k}$ as the edge from $v_i^j$ to $v_i^k$ with relation type $r_i^{j,k} \in \mathcal{R}_i$. The node features can be denoted as $\mathcal{F}_{\text{node}} = \{f_1^1, f_1^2, \ldots, f_N^M\} \in \mathbb{R}^{N \times M \times D}$, where $f_i^j \in \mathbb{R}^D$ represents the feature of node $v_i^j$ in the $j$-th frame. Here, $N$, $M$, and $D$ denote the number of frames in the video, the number of nodes per frame, and the dimensionality of each node feature, respectively.

## 3.2 Dynamic Prompt Module (DPM)

The DPM is a key component of our method. Since action disentanglement is driven by prompts, ambiguous prompts can make it hard for the model to focus on the specified action-relevant regions, leading to failure in disentangling the specified actions. we generate dynamic prompts based on the node features of the SSG, which guide the model to attend to action-relevant nodes and achieve better disentanglement. In this section, we introduce how prompts are generated based on arbitrary action specifications. We first describe the design of action specifications, followed by their construction strategy. Finally, we show how prompts are generated from the specifications, which guide the VGPNN to disentangle actions. Inspired by [8, 31], we inject learnable prompts into SSG feature space to accomplish guidance.

**Action Specification Design.** We first represent each action category as a one-hot vector. To specify a combination of actions to disentangle, we construct a multi-hot vector (e.g., $[1, 0, \ldots, 1]$), where each 1 indicates a hot element, i.e., a specified action. By aggregating several one-hot vectors, forming the Action Specification (AS). In designing the AS, we consider two components: one for the actions we intend to disentangle, called the Specified Action Prompt (SAP), and another for the actions we aim to exclude, called the Unspecified Action Prompt (UAP). For the design of SAP, a straightforward approach is to directly input the label-based SAP into the model. However, this can lead to shortcut learning, where the model relies on label cues rather than genuinely parsing the video content. Additionally, in many real-world scenarios, the exact set of actions in a video is not always known. To address these limitations, we further design an alternative version of SAP that includes both the ground-truth actions and distractor actions (i.e., actions not present in the video). This encourages the model to focus on learning discriminative features rather than simply following label hints. For the distractor-injected SAP design, we keep the number of hot elements in the SAP multi-hot vector to $K$, only some of which correspond to ground-truth actions, while the rest are distractors. To prevent the model from exploiting shortcuts, we construct the UAP by inverting the SAP. In this way, the combination of SAP and UAP covers the entire label set, ensuring a balanced and comprehensive specification of both present and absent actions (For better clarity, we include illustrative examples in Sec. C.1). In certain cases, SAP and UAP may contain only distractors (i.e., no true actions). For such scenarios, we introduce an auxiliary label called "no-action" to explicitly supervise the model.

**Action Specification Construction.** The AS consists of two types: distractor-injected and non-distractor-injected. The distractor-injected AS includes a combination of present labels (i.e., labels of some actions present in the video) and absent labels (i.e., labels of some actions absent in the video). Enumerating all possible label combinations is computationally infeasible. To address this, we adopt a random sampling strategy. For a video with $L$ ground-truth labels, the distractor-injected SAPs, we generate $L + 1$ different multi-hot vectors with a length $K$. Among them, $i$-th SAP contains $i$ random sampled present labels, and $K - i$ random sampled absent labels. In contrast, the non-distractor-injected SAPs, we generate $L$ different multi-hot vectors, only contains random sampled present labels. UAPs are then obtained by inverting each SAP. This results in $2L + 1$ different pairs of SAP and UAP. This design enables the model to learn from a diverse set of action combinations, promoting robust disentanglement by exposing it to both clean and noisy prompts while maintaining efficiency. For better clarity, we include illustrative examples in Sec. C.2.

**Prompt Generation.** A possible way to incorporate AS (symbolized as $\mathbf{v}$) is to project it into the same dimension space as the node features using a linear layer and add it directly to the features, we refer to this method as SimplePrompt. However, due to the static essence of such prompts, this

approach may struggle to capture contextual variations and risks encouraging shortcut learning, particularly in our dynamic disentanglement setting, where guidance need adapt to varying action contexts. To address this, we propose the Dynamic Prompt Module, which generates an adaptive prompt $p_i^j$ to each node feature $f_i^j$. The prompt $p_i^j$ is constructed as a linear combination of $T$ candidate prompts $\{q_1, q_2, \ldots, q_T\}$, which are generated from $\mathbf{v}$. The combination weights $\Delta \mathcal{P}^{i,j} = \{\alpha_1^{i,j}, \alpha_2^{i,j}, \ldots, \alpha_T^{i,j}\}$ are computed based on both $\mathbf{v}$ and $f_i^j$. It can be expressed as:

$$\Delta \mathcal{P}^{i,j} = \sigma(W^w \cdot (W^y \cdot \mathbf{v} + f_i^j)) \tag{1}$$

The resulting prompts is then added to $\mathcal{F}_{node}$ to generate prompted feature $\mathcal{F}_{node}^{\star}$, which replaces $\mathcal{F}_{node}$ in subsequent processing. It can be expressed as:

$$\mathcal{F}_{node}^{\star} = \{f_1^1 + p_1^1, f_1^2 + p_1^2, \ldots, f_N^M + p_N^M\} \qquad p_i^j = \sum_{t=1}^{T} \alpha_t^{i,j} \cdot q_t \tag{2}$$

### 3.3 Video Graph Parsing Neural Network (VGPNN)

We aim to disentangle subgraphs corresponding to specified actions. To achieve this, we require a network that, under the guidance of prompts, can parse action-relevant subgraphs by directionally propagating action-specific information. Inspired by GPNN [37], we design a variant for SSG, called VGPNN. GPNN parses action-relevant nodes by generating dynamic weights that control the direction of information flow. It consists of three key steps: Link, Message, and Update. In the Link step, it constructs edges between the same object nodes across different frames and learns aggregation weights for each timestamp. In the Message step, it multiplies these weights with messages, where each message is composed of a node's features and its edge features. In the Update step, it selects nodes from a specific frame as the initial hidden states, and updates them sequentially using the weighted messages from different time steps. However, GPNN has several limitations when applied to SSGs. 1)It only updates a node from a single frame, ignoring joint modeling of nodes across the entire video. 2) It performs shallow temporal modeling, as the learned weights are used only once for aggregation and are not further refined or propagated through deeper temporal structures.

**Video Graph Parsing Neural Network (VGPNN).** VGPNN is designed to disentangle features by decomposing a SSG into two subgraphs, guided by a given AS. To achieve selective information aggregation, with the same to GPNN [37], each VGPNN layer processes the entire SSG in three sequential steps. In Link step, we perform temporal modeling of edges $e^{j,k} = \{e_1^{j,k}, e_i^{j,k}, \ldots, e_N^{j,k}\}$ via self-attention [43] over the same interaction across time. In Message step, we generate dynamic weights from the edge features, which are then multiplied with the messages, where each message is composed of a node's features and its edge features. In Update step, we update all nodes by adding the weighted messages to the corresponding nodes directly. A MLP is then used to fuse the updated and original node features. More details please refer to Sec. B.

**Video Graph Normalization (VGNorm).** SSG exhibits characteristics of both graph structure and spatio-temporal properties of videos. However, many commonly used normalization methods tend to focus on only one aspect. For instance, BatchNorm [18] primarily normalizes across the temporal batch dimension, while LayerNorm [1] operates along the node (spatial) dimension to enforce spatial consistency. This one-sided focus limits network performance. To ensure temporal consistency and preserve the differences across SSGs from different videos, our method focuses on two key aspects. First, we use the frame-wise global mean $\mu_g \in \mathbb{R}^N$ to ensure temporal consistency across frames. During training, we compute the mean $\mu \in \mathbb{R}^N$ for each batch in real-time and update the $\mu_g \in \mathbb{R}^N$ using a momentum-based approach. Specifically, for batchsize $B$ nodes features $\mathcal{F}^{node} = \{\mathcal{F}_1^{node}, \mathcal{F}_2^{node}, \ldots, \mathcal{F}_B^{node}\} \in \mathbb{R}^{B \times N \times M \times D}$, where $\mathcal{F}_b^{node} \in \mathbb{R}^{N \times M \times D}$ is the $b$-th video nodes features, the global mean is computed as:

$$\mu_g := \alpha_m \cdot \mu_g + (1 - \alpha_m) \cdot \mu \qquad \mu = \frac{1}{B \cdot M \cdot D} \sum_{b=0}^{B} \sum_{j=0}^{M} \sum_{d=0}^{D} \mathcal{F}^{node} \tag{3}$$

where $\alpha_m$ represent the momentum. This global mean is used to normalize the features at each frame, ensuring temporal consistency.

Second, we introduce a mean scale shift $\alpha \in \mathbb{R}^N$ to account for the heterogeneity across different video SSGs. For the node features $\mathcal{F}_b^{node}$ of the $b$-th video, the normalization is defined as:

$$\mathcal{F}_b^* = \gamma \cdot \frac{\mathcal{F}_b^{node} - \alpha \cdot \mu_g}{\hat{\sigma}} + \beta \qquad \hat{\sigma}^2 = \frac{\sum_{j=0}^{M} \sum_{d=0}^{D} (\mathcal{F}_b^{node} - \alpha \cdot \mu_g)^2}{M \cdot D - 1} \tag{4}$$

Here, $\gamma, \beta \in \mathbb{R}^N$ are learnable affine parameters, and $\mathcal{F}_b^*$ is the normalized feature that replaces $\mathcal{F}_b$ in subsequent processing.

**Readout.** First, we compute dynamic weights for intra-frame nodes to aggregate them into frame-level features. Then, we calculate frame-level weights $s$ for each frame feature to perform temporal aggregation, yielding a global representation of the SSG, which is used for subsequent action classification supervision.

### 3.4 Optimization

In this section, we present the loss functions used to optimize our framework. Our design includes two complementary objectives: an **Action Classification Loss**, which guides the model to correctly disentangle features as specified by the AS, and an **Action Disentanglement Loss**, which ensures that the resulting representations are well separated. Together, these losses encourage both accurate action recognition and effective disentanglement in multi-action video scenes.

**Action Disentanglement Loss (AD Loss).** AD Loss includes a **reconstruction** loss and a **disentanglement** loss. For disentanglement, we treat the two disentangled representations as random variables and minimize their Pearson's correlation coefficient to enforce disentanglement. However, complete separation may discard shared semantics (e.g., common objects or interactions). To preserve these, we introduce a margin-based relaxation using a ReLU, allowing reasonable overlap while penalizing only excessive correlation. For a mini-batch with batchsize $B$, disentanglement loss $\mathcal{L}_{dis}$ can be expressed as:

$$\mathcal{L}_{dis} = \frac{1}{B} \cdot \sum_{b=0}^{B} \text{ReLU}\left(\frac{|\text{Cov}(\phi_b(\mathcal{F}_b^u), \psi_b(\mathcal{F}_b^s))|}{\sqrt{\text{Var}(\phi_b(\mathcal{F}_b^u))} \cdot \sqrt{\text{Var}(\psi_b(\mathcal{F}_b^s))}} - m_1\right) \tag{5}$$

where $\mathcal{F}_b^u, \mathcal{F}_b^s \in \mathbb{R}^{N \times M \times D}$ represent the action representations disentangled by UAP and SAP of $b$-th video, respectively. $\phi_b, \psi_b$ are measurable functions [14]. $\text{Cor}(\cdot), \text{Var}(\cdot), m_1$ represent covariance, variance and margin, respectively.

To ensure that the model performs disentanglement rather than feature extraction. We reconstruct the original feature $\mathcal{F}_b^o \in \mathbb{R}^{N \times M \times D}$, which is obtained from the video encoder before disentanglement. And we fuse the features $\mathcal{F}_b^u, \mathcal{F}_b^s$ and then obtain the reconstructed representations $\mathcal{F}_b^r \in \mathbb{R}^{N \times M \times D}$ with the reconstruction network. The reconstruction process can be expressed as

$$\mathcal{F}_b^r = \text{Net}_1(\delta \cdot \mathcal{F}_b^u + (1 - \delta) \cdot \mathcal{F}_b^s) \qquad \delta = \text{Net}_2([\mathcal{F}_b^u, \mathcal{F}_b^w]) \tag{6}$$

where $\text{Net}_1, \text{Net}_2$ represent reconstruction and fusion weight $\delta$ generation functions. And we apply a Mean Squared Error to enforce this constraint. Since explicit guidance is added. If we were to strictly align $\mathcal{F}_b^o$ with $\mathcal{F}_b^r$, the video encoder might focus on aligning with the prompts instead of the video. This could result in the video encoder extracting irrelevant information, causing the model to output results based on the prompts instead of the video. To prevent this, we reserve space by applying a margin-based ReLU relaxation $m_2$ to tolerate overlaps, reconstruction loss $\mathcal{L}_{rec}$ can be expressed as:

$$\mathcal{L}_{rec} = \frac{1}{B} \cdot \sum_{b=0}^{B} \text{ReLU}(\| \mathcal{F}_b^o - \mathcal{F}_b^r \|_2^2 - m_2) \tag{7}$$

**Overall.** In ProDA, we predict the labels of the disentangled actions specified by UAP and SAP, denoted as $a_u, a_s$ as well as the fused action label $a_t$ (i.e., the video label). The first two predictions ensure that the model correctly disentangles representations according to the AS, while the fused prediction encourages information completeness and promotes interaction between the disentangled components. During training, we have access to the ground-truth labels for UAP and SAP, denoted as $y_u, y_w$, along with the video ground-truth label $y_t$. Therefore, we can compute the total loss $\mathcal{L}$ as:

$$\mathcal{L} = \underbrace{\lambda_1 \cdot \mathcal{L}_{bce}(a_u, y_u) + \lambda_2 \cdot \mathcal{L}_{bce}(a_w, y_w) + \lambda_3 \cdot \mathcal{L}_{bce}(a_t, y_t)}_{\text{actions classification loss}} + \underbrace{\lambda_4 \cdot (\mathcal{L}_{dis} + \mathcal{L}_{rec})}_{\text{action disentanglement loss}} \tag{8}$$

where, $\mathcal{L}_{bce}$ is binary cross-entropy loss.

## 4 Experiments

### 4.1 Experiment Setup

**Dataset.** (1) The Charades dataset [40] consists of 9,848 videos, each with an average duration of 30 seconds. It covers 157 action categories, with each video containing an average of 6.8

Table 1: Multi-label action recognition performance comparison on the Charades's validation set in term of mAP. SSG:ground truth SSG. Bbox: bounding box from ground truth SSG.

| model | pretrain | Modality | mAP | model | pretrain | Modality | mAP |
|---|---|---|---|---|---|---|---|
| SGFB+NL [20] | K400 | RGB | 44.3 | SGFB+NL [20] | K400 | RGB+SSG | 60.3 |
| I3D+NL [4] | K400 | RGB | 37.5 | LaIAR [47] | K400 | RGB+Bbox | 63.6 |
| OR2G+NL [35] | K400 | RGB | 44.9 | OR2G [35] | K400 | RGB+SSG | 63.3 |
| LaIAR+NL [47] | K400 | RGB | 45.1 | LaIAR+NL [47] | K400 | RGB+Bbox | 67.4 |
| VicTR(B/16) [23] | CLIP | RGB | 50.1 | OR2G+NL [35] | K400 | RGB+SSG | 67.5 |
| VicTR(L/14) [23] | CLIP | RGB | 57.6 | **Ours(B/16)** | **CLIP** | **RGB+SSG** | **71.1** |

Table 2: Human-Human Interaction Classification results in SportsHHI. SportsHHI$^\star$ is our reimplementation based on the official SportsHHI code, with minor modifications (e.g., parameter size) to match the capacity of ProDA for fair comparison.

| method | mAP | R@150 | R@100 | R@50 | R@20 | Params |
|---|---|---|---|---|---|---|
| STTran [5] | 3.31 | 71.29 | 62.91 | 42.67 | 22.14 | - |
| HORT [19] | 3.75 | 78.57 | 67.78 | 50.33 | 26.96 | - |
| SlowFast [11] | 5.00 | 81.66 | 74.00 | 52.74 | 26.82 | - |
| ACARN [36] | 5.44 | 82.85 | 75.22 | 56.53 | 31.77 | - |
| SportHHI [50] (baseline) | 10.69 | 89.25 | 82.93 | 68.13 | 43.72 | 111.18M |
| SportsHHI$^\star$ (reimpletmented) | 10.16 ($\downarrow$0.53) | 90.07 ($\uparrow$0.82) | 84.83 ($\uparrow$1.9) | 68.40 ($\uparrow$0.37) | 40.16 ($\downarrow$3.56) | 123.11M |
| **Ours (baseline+ProDA)** | **13.55 ($\uparrow$2.86)** | **92.54 ($\uparrow$3.29)** | **87.91 ($\uparrow$4.98)** | **72.69 ($\uparrow$4.56)** | **47.81 ($\uparrow$4.09)** | 123.19M |

distinct actions. Multiple actions can occur simultaneously, making the recognition task particularly challenging. Based on Charades, the Action Genome dataset [20] provides fine-grained annotations by decomposing actions and focusing on video clips where the actions take place. It contains 234K keyframes, with annotations for 476K object bounding boxes and 1.72 million object relationships. (2) The SportsHHI dataset [50] is a specially designed dataset focusing on human-human interaction in sports. It covers basketball and volleyball games, and video data are selected from MultiSports [28]. This dataset labels 34 interaction classes with 118,075 human bounding boxes and 50,649 interaction instances. Following [47, 50], we conduct evaluations on Charades for multi-label action recognition and on SportsHHI for Human-Human interaction Classification (HHICls), adopting mean Average Precision (mAP) as the metric.

**Experiment Setup.** (1) On Charades, all experiments are conducted under the same design. For each video, the network takes $N$ frames as input. During training, frames are randomly sampled from the video, while during validation, the frames are extracted evenly over the whole video. In the first stage, the entire network is trained end-to-end, and the resulting weights are used to initialize the second stage. In the second stage, only the classifiers for $a_u, a_s, a_t$ are trained. Additionally, to enable deeper analysis, we introduce an auxiliary prediction $a_m$, which is inferred from the features before disentanglement (features from Video Encoder as illustrated in Figure 2). Its corresponding classifier is also trained during the second stage. (2) On SportsHHI, we adopt the official open-source model and training code without altering any training settings. For fair evaluation, we conduct two experiments: 1) extending the official baseline with our proposed modules, namely the DPM and AD Loss; and 2) constructing a parameter-matched baseline by enlarging its model size to match ours, without incorporating any proposed components.

## 4.2 Compared with State-of-The Art Methods

**Charades.** We compare the action recognition accuracy of the proposed method and the State-of-The-Art methods (SoTA) on the Charades. Table 1 presents our results on the Charades dataset. ProDA achieves state-of-the-art performance. Our method outperforms several CNN- and ViT-based approaches, including I3D [4] and VicTR [23]. This demonstrates that our approach can better capture object interactions within structured data (e.g., SSG), enabling a deeper understanding of video. Notably, our method with a CLIP-B/16 [38] still surpasses VicTR (L/14) [23], even though we only use an image encoder, while VicTR [23] additionally employs a text encoder. Furthermore, our method also outperforms several approaches that leverage SSG [35, 47]. This demonstrates that explicitly disentangling and processing actions separately leads to a more accurate understanding of complex actions. It is also worth noting that our method achieves this performance without requiring any additional external models.

Table 3: Ablation of prompt

| prompt type | $a_t$mAP (%) | $a_m$mAP (%) |
|---|---|---|
| SimplePrompt | 68.07 | 57.80 |
| DPM | **71.10** | **59.04** |

Table 4: Ablation of normalization

| norm type | $a_t$mAP (%) | $a_m$mAP (%) |
|---|---|---|
| LayerNorm [1] | 68.12 | 57.96 |
| BatchNorm [18] | 65.39 | 58.20 |
| GraphNorm [3] | 68.61 | 58.32 |
| VGNorm | **71.10** | **59.04** |

Table 5: Ablation study of loss function on Charades dataset. ❻ denotes the baseline training only the video encoder without ProDA; ❶ denotes using ProDA with only the action classification loss.

| | $\mathcal{L}_{res}$ w/o margin | $\mathcal{L}_{res}$ | $\mathcal{L}_{dis}$ w/o margin | $\mathcal{L}_{dis}$ | $a_t$ mAP on Charades (%) | $a_m$ mAP on Charades (%) |
|---|---|---|---|---|---|---|
| ❶ | - | - | - | - | 67.45 | 57.38 |
| ❷ | ✓ | - | ✓ | - | 68.43 | 8.03 |
| ❸ | - | - | - | ✓ | 67.78 | 58.28 |
| ❹ | - | ✓ | - | - | 64.89 | 57.90 |
| ❺ | - | ✓ | - | ✓ | **71.10** | **59.04** |
| ❻ | - | - | - | - | - | 58.01 |

**SportsHHI.** We compare our method with the SoTA baseline and its parameter-increased variant, as reported in Table 2. Enlarging the baseline leads to a 10.8% increase in parameters, whereas our proposed DPM introduces only 1.46M additional parameters (1.18% of the total). Despite such a small overhead, our method achieves notable improvements across all metrics. Even under comparable parameter settings, it consistently outperforms the baseline. These results demonstrate the strong generalization capability of our approach and confirm that the performance gain primarily arises from the proposed AD Loss rather than from increased model capacity.

## 4.3 Ablation Study

**Impact of DPM.** DPM is a key component of our framework. To verify its effectiveness, we conduct comparisons with both Simple Prompt and DPM. In addition to evaluating the final fused output $a_t$, we also train a separate classifier on $a_m$, the features before disentanglement, for a more comprehensive analysis. The results in Table 3 show that using DPM improves $a_t$ by 3.03% mAP and $a_m$ by 1.24% mAP compared to the SimplePrompt. These findings indicate that our dynamically generated prompts, conditioned on the input, better capture the evolving nature of actions in videos, enabling more effective representation disentanglement. Furthermore, the clearer disentangled features also enhance the performance of the video encoder network.

**Impact of VGNorm.** We design VGNorm to adapt to SSG, which integrate both video and graph structures. We compare it with several widely used normalization methods, and observe that VGNorm achieves the best performance as shown in Table 4. Compared to BatchNorm [18], VGNorm better preserves the spatial structure of graphs while maintaining temporal consistency, leading to a significant improvement of +5.71% mAP. Compared to LayerNorm [1], which normalizes features independently and ignores the graph structure, VGNorm incorporates spatially consistent mean scaling to preserve temporal consistency and capture the heterogeneity across SSGs from different videos, resulting in a +2.98% mAP improvement. Finally, compared to GraphNorm [3], VGNorm enhances temporal consistency, enabling the model to better capture temporal dynamics, which contributes to a +2.49% mAP improvement. Moreover, by jointly modeling both temporal consistency and the heterogeneity across SSGs, VGNorm produces more stable and adaptive representations. This leads to the best performance on $a_m$, further demonstrating the effectiveness of our normalization strategy.

**Impact of AD Loss.** We design ADLoss to enable effective feature disentanglement by introducing a margin-based reconstruction loss and a disentanglement loss. As shown in Table 5, (❶ *v.s.* ❷) removing the margin constraint results in higher $a_t$ performance, but the accuracy of $a_m$ drops sharply to 8.03% mAP, indicating that the video encoder fails to retain meaningful representations. This suggests that, without the margin, the model learns to take a shortcut by relying solely on the prompt to generate features, bypassing true video understanding. In contrast, when only disentanglement is applied (❶ *v.s.* ❸), the model still achieves notable improvements, demonstrating that action disentanglement alone can help capture object states and interaction transitions more effectively. Comparison between (❶ *v.s.* ❹) further shows that treating all actions as a whole leads to suboptimal results, as the model cannot preserve the independence of different actions. However, the disentanglement structure under reconstruction constraints pushes the video encoder to extract richer features, thereby improving its performance. Moreover, (❸ *v.s.* ❺) shows that adding reconstruction transforms the process from mere feature extraction to genuine disentanglement, preserving more interaction cues. Finally, (❹ *v.s.* ❺) confirms that combining reconstruction with disentanglement ensures the independence of action representations, allowing the model to more accurately capture the dynamic interactions within video, ultimately achieving superior performance.

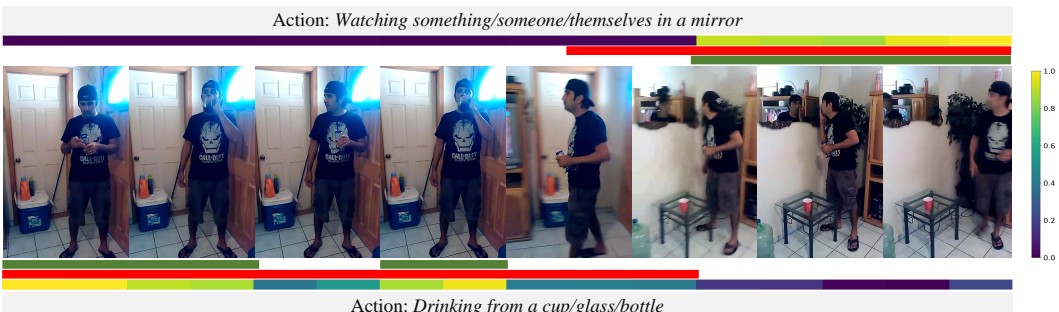

Figure 3: Frame-level weights $s$ predicted by the readout function for different non-distractor-injected SAPs within the same video. The heatmaps (*colored bars*) visualize the predicted weights, reflecting the relevance of each frame to the target action. The *red bars* indicate the ground-truth action segments, while the *green bars* denote our localization results obtained by thresholding $s$ at 0.7.

Table 6: Non-distractor-injected SAP. label nums indicates the length of the ground-truth label set for each video

Table 7: Distractor-injected SAP. label nums indicates the length of the ground-truth label set for each video

| label nums | $a_s$ mAP | label nums | $a_s$ mAP | label nums | $a_s$ mAP | label nums | $a_s$ mAP |
|---|---|---|---|---|---|---|---|
| 1 | 100.00 ±0.0 | 9 | 99.80 ±0.3 | 1 | 100.0 ±0.0 | 9 | 85.63 ±6.6 |
| 2 | 100.00 ±0.0 | 10 | 99.45 ±0.6 | 2 | 90.52 ±3.5 | 10 | 85.87 ±6.7 |
| 3 | 100.00 ±0.0 | 11 | 99.62 ±0.4 | 3 | 92.61 ±1.1 | 11 | 86.56 ±5.0 |
| 4 | 99.66 ±0.4 | 12 | 99.10 ±1.0 | 4 | 93.16 ±0.7 | 12 | 84.85 ±6.5 |
| 5 | 99.80 ±0.2 | 13 | 99.56 ±0.6 | 5 | 87.17 ±4.8 | 13 | 87.42 ±6.4 |
| 6 | 99.82 ±0.3 | 14 | 98.98 ±1.1 | 6 | 87.69 ±2.9 | 14 | 87.73 ±6.0 |
| 7 | 99.98 ±0.0 | 15 | 99.19 ±1.2 | 7 | 86.65 ±6.1 | 15 | 88.58 ±6.0 |
| 8 | 99.88 ±0.1 | 16 | 99.80 ±0.4 | 8 | 86.32 ±6.1 | 16 | 92.10 ±4.9 |

## 4.4 Further Analysis

**Disentanglement Robustness Analysis.** To evaluate the robustness of our action disentanglement, we analyze the $a_s$ accuracy of videos with different numbers of ground-truth labels (denoted as label nums in Table 6 7) when exposed to varying numbers of present labels in the SAP, under both distractor-injected and non-distractor settings. As shown in Table 7, in the distractor-injected case (SAP contains the labels not present in video), we observe a non-monotonic trend: accuracy drops when the number of present labels increases from 5 to 12, but rises again as present labels dominate the input. We attribute the initial drop to the increasing complexity of the prediction task due to more competing present labels under noisy input. The subsequent performance increase is attributed to reduced noise as more present labels are included. Despite the precision varying between 85% and 100% under distractor-injected, our model demonstrates robust performance across different input conditions. Even in the worst-case scenario, where the precision is at its lowest (85%), the model still maintains a relatively high accuracy. This indicates its ability to handle incomplete or noisy data effectively. Moreover, in the non-distractor-injected case (SAPs only contain labels present in video), as shown in Table 6, the model achieves nearly perfect accuracy, reaching close to 100%. This shows that in ideal conditions, the model is able to fully exploit the available information. The robustness in noisy conditions and strong performance on clean data indicate that our model generalizes well and benefits from mutual reinforcement between the two settings.

**Temporal Action Localization.** Although our method does not explicitly design a localization module, the disentangled representation naturally provides cues for action localization. Specifically, within our readout function, we aggregate temporal features using learned frame-level weights $s$. These weights reflect the relevance of each frame to the target action, and can thus be interpreted as frame-level localization scores. By normalizing these weights, we obtain a per-frame prediction that serves as our localization output. To obtain frame-level ground truth for localization, we project the action segment annotations onto frames: all frames within the annotated action duration are labeled as positive (1), and the rest as negative (0). We binarize the normalized weights $s$ using a threshold $\theta$, assigning 1 to frames with $s > \theta$ and 0 otherwise, and evaluate the resulting predictions using frame-level mAP based on $a_s$. As shown in Table 8 9, we observe that the variation in average precision across different $\theta$ values is larger than the variation across different IoU thresholds, and the gap between distractor-injected and non-distractor-injected settings narrows as $\theta$ increases. This

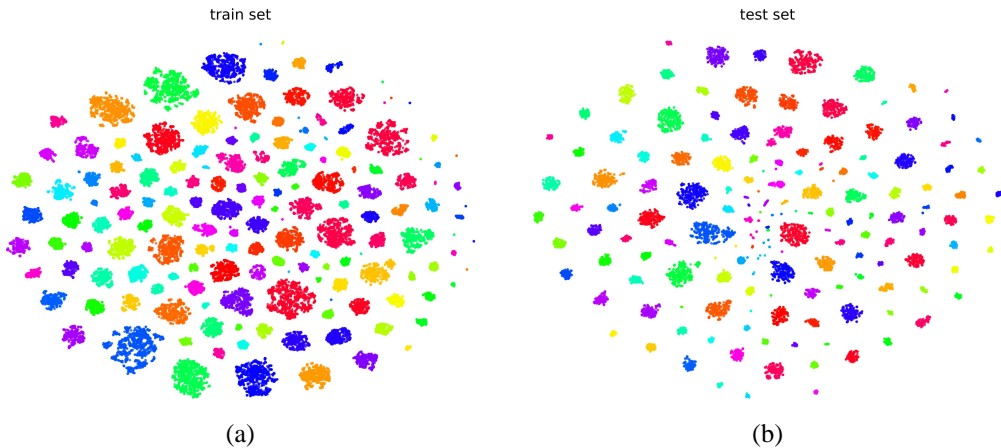

Figure 4: t-SNE visualization of disentangled action features across 157 classes on Charades [40]. (a) Training set. (b) Test set.

Table 8: Distractor-injected SAP

|  |  | IoU threshold | | | |
|---|---|---|---|---|---|
|  |  | 0.2 | 0.5 | 0.7 | **Avg.** |
| | 0.2 | 77.97 | 43.68 | 26.08 | 49.24 |
| $\theta$ | 0.5 | 64.25 | 33.70 | 20.78 | 39.57 |
| | 0.7 | 53.80 | 22.98 | 12.99 | 29.92 |
| **Avg.** | | 65.34 | 33.45 | 19.95 | - |

Table 9: Non-distractor-injected SAP

|  |  | IoU threshold | | | |
|---|---|---|---|---|---|
|  |  | 0.2 | 0.5 | 0.7 | **Avg.** |
| | 0.2 | 85.56 | 57.44 | 40.18 | 61.06 |
| $\theta$ | 0.5 | 74.44 | 43.43 | 27.75 | 48.53 |
| | 0.7 | 59.27 | 24.26 | 13.14 | 32.23 |
| **Avg.** | | 73.09 | 41.71 | 27.03 | - |

indicates that the introduced noise (i.e., absent labels) mainly adds irrelevant information without significantly disrupting the separation of true actions, demonstrating the robustness of our method. In the non-distractor-injected setting, the model achieves notably better localization performance, confirming that the high recognition accuracy is not due to shortcut learning, but rather due to effective utilization of SAP information for representation disentanglement under clean inputs.

### 4.5 Visualization

We computed the inter-class and intra-class distances after applying t-SNE [32] for dimensionality reduction on Charades [40], with the visualization results shown in Figure 4. Specifically, for each video with $L$ labels, we disentangle $L$ action features, covering a total of 157 action classes. We further calculate the inter-class and intra-class distances on both the training and test sets. For the training set, the intra-class and inter-class distances are 2.08 ± 1.57 and 58.58 ± 27.99, respectively, while for the test set they are 1.79 ± 1.45 and 74.94 ± 36.98. Notably, the intra-class distances of disentangled features are mostly within 1, whereas the inter-class distances remain substantially larger. This demonstrates that our feature disentanglement effectively increases the separability between action categories while enhancing the compactness of features within the same category, thereby providing more discriminative representations for downstream action recognition and understanding. We provide more visualization results in Sec E.

## 5 Conclusion

In this paper, we propose Prompt-guided Representation Disentanglement for Action Recognition (ProDA), which is capable of disentangling any complete action from Spatio-temporal Scene Graphs (SSGs). ProDA leverages SSGs and introduces Dynamic Prompt Module (DPM) to guide a Graph Parsing Neural Network (GPNN) in extracting action-specific representations. We further design a video-adapted GPNN that aggregates information using dynamic weights. To enhance performance, we introduce VGNorm and AD Loss, which ensure the spatio-temporal consistency across SSGs of different videos and the completeness of feature disentanglement, respectively. Experimental results demonstrate the superiority of our method.

## Acknowledgments and Disclosure of Funding

This work was supported by grants from the Natural Science Foundation of Shanxi Province (2024JCJCQN-66), Science and Technology Commission of Shanghai Municipality (NO.24511106900), Key R&D Program of Zhejiang (2024SSYS0091).

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

# A  Video Encoder

We adopt Dual-AI [15] as our video encoder, which is a two-stream architecture composed of interleaved temporal and spatial Transformers [7]. This design facilitates effective spatio-temporal information exchange through alternate modeling along temporal and spatial dimensions. On top of the original architecture, we further introduce additional spatial-to-spatial and temporal-to-temporal branches to enhance the flow of information across both dimensions. The overall architecture is illustrated in Figure 5. After the video encoder, an Multi-Layer Perceptron (MLP) fuses the two branches, and a Feed Forward Network (FFN) [43] further integrates the features to produce the input $\mathcal{F}_o^f$ for Video Graph Parsing Neural Network (VGPNN).

Both the Spatial Transformer and the Temporal Transformer adopt the standard self-attention encoder architecture [43] (as shown in Figure 6), with the primary difference lying in the dimension along which the attention is applied in spatial or temporal dimension, as shown in Figure 7.

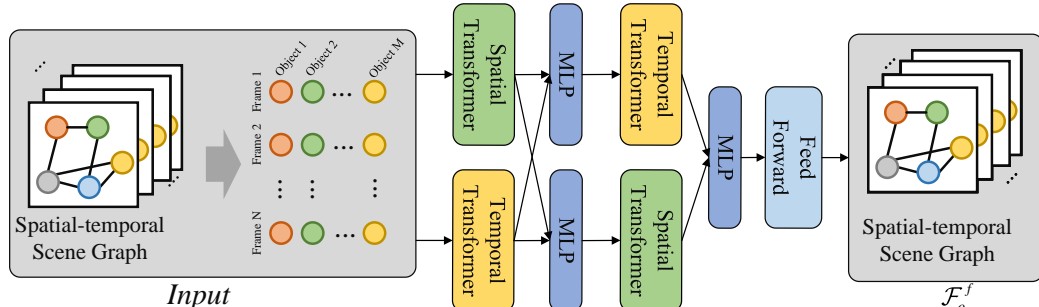

Figure 5: Our video encoder builds upon Dual-AI [15] by introducing additional spatial-spatial and temporal-temporal streams.

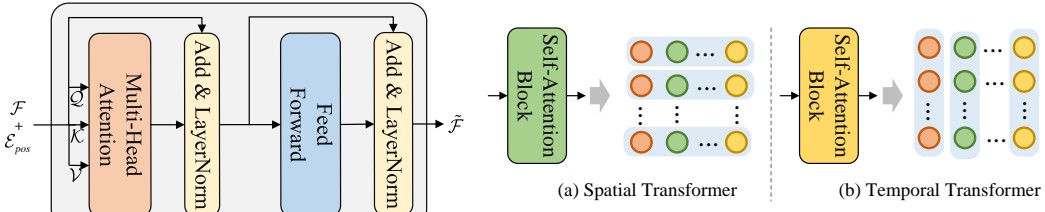

Figure 6: Architecture of self-attention encoder.

Figure 7: Spatial Transformer and Temporal Transformer

# B  Video Graph Parsing Neural Network

## B.1  Formulation

In SSG, each frame of a video is processed into a graph, where nodes represent the objects present in that frame, and edges represent the semantic relationships between these objects (e.g., spatial or interaction relations). Formally, let $\mathcal{G} = (\mathcal{V}, \mathcal{E}, \mathcal{R})$ denote the complete SSG constructed from the video, where $\mathcal{V}$ is the set of all object nodes across all frames, $\mathcal{E}$ is the set of all directed edges, and $\mathcal{R}$ is the set of all relation types. Each node $v \in \mathcal{V}$ is uniquely indexed from $\{1, \ldots, |\mathcal{V}|\}$. Each edge $e = (v, w, r) \in \mathcal{V} \times \mathcal{V}$ represents a directed semantic relation of type $r \in \mathcal{R}$ from node $v$ to node $w$. Let $g_i = (\mathcal{V}_i, \mathcal{E}_i, \mathcal{R}_i)$ denote the scene graph of the $i$-th frame, where $\mathcal{V}_i \subseteq \mathcal{V}$, $\mathcal{E}_i \subseteq \mathcal{E}$ and $\mathcal{R}_i \subseteq \mathcal{R}$ denote the frame-specific object nodes, their edges and relations. We denote $v_i^j$ as the $j$-th object node in frame $i$, and $e_i^{j,k}$ as the edge from $v_i^j$ to $v_i^k$ with relation type $r_i^{j,k} \in \mathcal{R}_i$. The node features can be denoted as $\mathcal{F}_{\text{node}} = \{f_1^1, f_1^2, \ldots, f_N^M\} \in \mathbb{R}^{N \times M \times D}$, where $f_i^j \in \mathbb{R}^D$ represents the feature of node $v_i^j$ in the $j$-th frame. Here, $N$, $M$, and $D$ denote the number of frames in the video, the number of nodes per frame, and the dimensionality of each node feature, respectively.

It is important to note that the number of nodes in each frame is the same. The $i$-th node in every frame corresponds to the same object, though its visual features may vary across frames. This ensures that each node represents the same object across frames, even though its appearance changes. However, not every object appears in every frame, leading to padding where necessary.

**Link.** For edge $e_i^{j,k}$, we have $N$ triples across frames, denoted as $\{(v_1^j, v_1^k, r_1^{j,k}), \ldots, (v_N^j, v_N^k, r_N^{j,k})\}$. To model the transition of object interactions over time, we first extract interaction features from these edge triples. Specifically, we concatenate the components of each edge triple and feed them into an edge function $\texttt{Edge}(\cdot)$ (implemented using a Multi-Layer Perceptron, MLPs; unless otherwise specified, all functions in subsequent modules are implemented as MLPs) to obtain frame-wise relation features $f^{j,k} = \{f_1^{j,k}, f_2^{j,k}, \ldots, f_N^{j,k}\}$. We then apply a self-attention encoder [43] to model the dependencies and transitions among these interaction features across frames to obtain interaction feature $\mathcal{F}^{j,k} = \{\mathcal{F}_1^{j,k}, \mathcal{F}_2^{j,k}, \ldots, \mathcal{F}_N^{j,k}\}$ as follow:

$$\mathcal{F}^{j,k} = \texttt{SelfAttEncoder}(\underbrace{\texttt{Edge}([v_1^j, v_1^k, r_1^{j,k}]), \ldots, ([v_N^j, v_N^k, r_N^{j,k}])}_{\text{relation features } f^{j,k}}) \tag{9}$$

where $[\cdot]$ represent concatenation, and $\texttt{SelfAttEncoder}(\cdot)$ is the self-attention encoder with $L$ standard self-attention encoder blocks (as shown in Figure 6).

**Message & Update.** To facilitate better disentanglement, we employ dynamic weights to modulate the information propagated through different edges and nodes. This not only enables selective information flow, but also provides interpretability by highlighting the importance of each interaction. We through interaction feature $\mathcal{F}^{j,k}$ to obtain weight feature $\mathcal{F}_w^{j,k}$ as follows:

$$\mathcal{F}_w^{j,k} = \sigma(W^m \cdot \mathcal{F}^{j,k}) \tag{10}$$

where $W^m$ denotes a linear transformation projecting to a scalar. We then combine the interaction features and node features to compute the messages $\mathcal{F}_m^j$ to be passed to node $v^j = \{v_1^j, v_2^j, \ldots, v_N^j\}$ in each frame, which are dynamically aggregated using the weight features $\mathcal{F}_w^j$, as follows:

$$\mathcal{F}_m^j = \sum_{k=0}^{K} \left(\mathcal{F}_w^{j,k} \cdot \texttt{Msg}([v^j, \mathcal{F}^{j,k}])\right) \tag{11}$$

where $K$ denotes the number of neighbors connected to node $v^j$, and $\texttt{Msg}(\cdot)$ represents the message function. After computing the message features $\mathcal{F}_m^j$, we use an aggregation function to incorporate them into the updated representation of node $v_j$, as follows:

$$\hat{v^j} = \texttt{Agg}(v^j \oplus \mathcal{F}_m^j) \tag{12}$$

where $\hat{v^j}$, $\texttt{Agg}(\cdot)$, and $\oplus$ denote updated node representation, the aggregation function, and element-wise addition, respectively. $\hat{v^j}$ replaces $v^j$ in following processing.

Compared to previous work Graph Parsing Neural Network (GPNN) [37], VPGNN introduces stronger temporal modeling capabilities by modeling edges over time, rather than just generating aggregation weights for edges. VGPNN updates node features at each layer, while GPNN primarily updates edges and only updates nodes in the final layer. Additionally, VGPNN retains the full SSG across frames, unlike GPNN, which only preserves nodes from a single frame. This results in VGPNN offering better interpretability and generalization.

## C  Action Specification

### C.1  Design

We assume that the label length of a dataset is 5. For a video labeled with classes $\{0, 1\}$, its multi-hot representation is:

$$[1, 1, 0, 0, 0] \tag{13}$$

Then, there will be three Specified Action Prompts (SAPs), corresponding to $\{0, 1\}$, $\{0\}$, and $\{1\}$, respectively (As shown in Eq. 14).

$$\begin{aligned} \{0, 1\} &\to [1, 1, 0, 0, 0], \\ \{0\} &\to [1, 0, 0, 0, 0], \\ \{1\} &\to [0, 1, 0, 0, 0] \end{aligned} \tag{14}$$

However, using only the SAP may lead the model to take shortcuts, focusing solely on the SAP rather than the video content. Therefore, we augment the SAP with distractor classes that are not associated with the video. In this case, the distractor set is $\{2, 3, 4\}$. Assuming **K=3**, we take the second example in Equation 14 (i.e., with label $\{0\}$)to illustrate: it will generate three **distractor-injected SAPs** (As shown in Eq. 15).

$$
\begin{aligned}
&[1, 0, 1, 1, 0], \\
&[1, 0, 1, 0, 1], \\
&[1, 0, 0, 1, 1]
\end{aligned}
\tag{15}
$$

Similarly, taking the first example in Eq 14 (i.e., with label $\{0, 1\}$), with label, it will also generate three **distractor-injected SAPs** (As shown in Eq 16).

$$
\begin{aligned}
&[1, 1, 1, 0, 0], \\
&[1, 1, 0, 1, 0], \\
&[1, 1, 0, 0, 1]
\end{aligned}
\tag{16}
$$

In both examples, the number of ones in the multi-hot vector is 3.

We divide the features into two parts: one corresponding to the specified action and the other to the remaining actions. If we feed the SAP into the part that is intended for unrelated actions, it will lead to information leakage, since the SAP contains the ground-truth labels. To avoid this, we instead feed the full label space of the dataset. Specifically, we obtain the Unspecified Action Prompts (UAP) by taking the complement of the SAP with respect to the entire label space, such that the combination of SAP and UAP covers all action categories in the dataset.

Taking Eq. 15 as an example, its SAP is given by:

$$
\begin{aligned}
&[0, 1, 0, 0, 1], \\
&[0, 1, 0, 1, 0], \\
&[0, 1, 1, 0, 0]
\end{aligned}
\tag{17}
$$

For Eq. 16, its SAP is given by:

$$
\begin{aligned}
&[0, 0, 0, 1, 1], \\
&[0, 0, 1, 0, 1], \\
&[0, 0, 1, 1, 0]
\end{aligned}
\tag{18}
$$

### C.2 Construction

The AS (SAP and UAP pair) consists of two types: distractor-injected and non-distractor-injected. The distractor-injected AS includes a combination of present labels (i.e., labels of some actions in groundtruth labels) and absent labels (i.e., labels of some actions absent in the groundtruth labels), while non-distractor-injected only contains present labels.

For a video with the ground truth label set {Putting a pillow somewhere, Taking a pillow from somewhere, Throwing a pillow somewhere, Holding a pillow, Tidying up a blanket/s} corresponding to label indices {77,79,80,76,75}, and **presents labels are sample from this ground truth label set**. The remaining labels, i.e., all other labels from the full set $\{0, 1, \ldots, 156\}$ excluding the ground truth labels. **Absent labels are sampled from remaining labels.**

Under the non-distractor-injected setting, we generate 5 SAPs:

$$
\begin{aligned}
&[77], \\
&[80, 76], \\
&[77, 80, 75], \\
&[79, 77, 76, 75], \\
&[80, 77, 76, 75, 79]
\end{aligned}
\tag{19}
$$

Each SAP contains only present labels, with 1 to 5 ground truth labels randomly sampled from the ground truth label set. To prevent information leakage, **we use all labels outside each SAP as the**

**corresponding UAP**, thereby disentangling the rest of actions. Consequently, for the 5 SAPs, we generate 5 corresponding UAPs.

$$
\begin{aligned}
&\text{4 present labels: } [\underbrace{\textbf{80,76,75,79}, 0, 1, \ldots, 156}_{156 \text{ labels}}], \\
&\text{3 present labels: } [\underbrace{\textbf{77,79,75}, 0, 1, \ldots, 156}_{155 \text{ labels}}], \\
&\text{2 present labels: } [\underbrace{\textbf{79,76}, 0, 1, \ldots, 156}_{154 \text{ labels}}], \\
&\text{1 present labels: } [\underbrace{\textbf{80}, 0, 1, \ldots, 156}_{153 \text{ labels}}], \\
&\text{0 present labels: } [\underbrace{0, 1, \ldots, 156}_{152 \text{ labels}}]
\end{aligned}
\tag{20}
$$

Under the distractor-injected setting, **we fix the length of each SAP to 16**, and generate $5 + 1 = 6$ SAPs:

$$
\begin{aligned}
&\text{0 present labels:} [152, 32, 24, 125, 145, 110, 93, 138, 129, 44, 124, 130, 85, 7, 5, 0], \\
&\text{1 present labels:} [\textbf{76}, 143, 111, 19, 104, 23, 112, 12, 78, 146, 56, 136, 116, 9, 24, 125], \\
&\text{2 present labels:} [\textbf{77, 76}, 140, 111, 134, 130, 23, 18, 155, 136, 88, 133, 112, 127, 17, 2], \\
&\text{3 present labels:} [\textbf{76, 79, 75}, 47, 125, 91, 71, 39, 66, 141, 9, 27, 52, 1, 132, 54], \\
&\text{4 present labels:} [\textbf{76, 79, 75, 77}, 56, 88, 120, 22, 128, 62, 52, 105, 142, 124, 153, 35] \\
&\text{5 present labels:} [\textbf{80, 77, 76, 75, 79}, 17, 51, 109, 143, 103, 57, 15, 152, 130, 125, 47]
\end{aligned}
\tag{21}
$$

Each SAP contains both present and absent labels, with 0 to 5 ground truth labels randomly sampled from the original ground truth label set, and the rest 16 to 11 labels sampled from the remaining labels. To prevent information leakage, **we use all labels other than those in each SAP as the corresponding UAP**, thereby disentangling the other actions. **Each UAP has a length of 141**. Consequently, for the five SAPs, we generate 6 corresponding UAPs, each of length 141.

$$
\begin{aligned}
&\text{5 present labels: } [\underbrace{\textbf{80,76,77,75,79}, 1, 2, \ldots, 156}_{141 \text{ labels}}], \\
&\text{4 present labels: } [\underbrace{\textbf{80,77,75,79}, 0, 1, \ldots, 156}_{141 \text{ labels}}], \\
&\text{3 present labels: } [\underbrace{\textbf{80,79,75}, 0, 1, \ldots, 156}_{141 \text{ labels}}], \\
&\text{2 present labels: } [\underbrace{\textbf{80,77}, 0, 2, \ldots, 156}_{141 \text{ labels}}], \\
&\text{1 present labels: } [\underbrace{\textbf{80}, 0, 1, \ldots, 156}_{141 \text{ labels}}], \\
&\text{0 present labels: } [\underbrace{0, 1, \ldots, 156}_{141 \text{ labels}}]
\end{aligned}
\tag{22}
$$

Finally, for a video with 5 labels, we generate $2 \cdot 5 + 1 = 11$ ASs (11 SAP and UAP pairs).

Table 10: This table outlines the division of the Charades dataset into five subsets, ensuring that there is no overlap between the scenes used for training and testing

| Subdataset | Training Scene | Test Scene |
|---|---|---|
| Scenario1 | Stairs,Laundry room,Home Office, Hallway,Bedroom,Pantry,Dining room,Entryway | Living room,Closet,Kitchen,Bathroom, Garage,Recreation room,Basement,Other |
| Scenario2 | Laundry room,Bathroom,Pantry,Closet, Entryway,Recreation room,Garage,Other | Bedroom,Living room,Kitchen,Home Office, Hallway,Stairs,Basement,Dining room |
| Scenario3 | Stairs,Laundry room,Bedroom,Basement, Bathroom,Entryway,Recreation room,Other | Living room,Closet,Kitchen,Home Office, Garage,Hallway,Pantry,Dining room |
| Scenario4 | Kitchen,Stairs,Laundry room,Home Office, Bedroom,Bathroom,Pantry,Dining room | Living room,Closet,Garage,Hallway, Recreation room,Entryway,Basement,Other |
| Scenario5 | Kitchen,Laundry room,Hallway,Basement, Dining room,Living room,Closet,Other | Bedroom,Home Office,Bathroom,Garage, Stairs,Recreation room,Entryway,Pantry |

Table 11: Domain shift experiment on the Charades dataset

| Method | mAP (%) |
|---|---|
| STIGPN [46] | 54.1 |
| LaIAR [47] | 57.2 |
| Ours | **63.0** |

# D   Details and More Results for Our Experiment

## D.1   Additional Experiment

We conduct domain shift experiments on the Charades dataset to evaluate the generalization ability of our method. Specifically, we follow the experimental setup of [47], splitting Charades into five disjoint subsets with different action groups. The details are shown in Table 10.

As shown in Table 11, our method achieves the best performance under domain shift settings, indicating strong generalization ability across different scenarios. We attribute this to the effective disentanglement of features, which allows the model to better capture action-relevant information, thereby improving action understanding and enhancing generalization performance in unseen environments.

## D.2   More Details

For each video, we uniformly sample 16 frames as input. We adopt a two-stage training strategy for all experiments. The first stage (pre-train) runs for 5 epochs with a learning rate of 2e-4, aiming to learn the representations for $a_u$ and $a_s$. In the second stage (post-train), we load the checkpoint from the epoch with the best performance on $a_u$ and $a_s$ in the pre-training stage, and only train the classifier for $a_t$. This stage is also trained for 5 epochs with a learning rate of 1e-4. We report the best-performing epoch from the second stage. The detailed results are shown in Table 12 13. Except for our method, other approaches generally show a certain degree of performance improvement across different settings. Notably, our method consistently achieves the best performance in all scenarios, demonstrating its robustness and superior generalization capability.

# E   Visualization

Figure 8 shows the action disentanglement localization results for two different videos (with 3 and 4 action labels respectively). Both videos contain concurrent or individual interactions. Figure 9 illustrates the process by which our method disentangles the specified actions from the concurrent interactions.

Table 12: Ablation of normalization

| norm type | Post-train $a_t$mAP (%) | Pre-train $a_t$mAP (%) |
|---|---|---|
| LayerNorm [1] | 68.12 | 65.44 |
| BatchNorm [18] | 65.39 | 63.05 |
| GraphNorm [3] | 68.61 | 65.32 |
| VGNorm | **71.10** | **71.32** |

Table 13: Ablation study of loss function on Charades dataset. ❶ denotes using ProDA with only the action classification loss.

| | $\mathcal{L}_{res}$ w/o margin | $\mathcal{L}_{res}$ | $\mathcal{L}_{dis}$ w/o margin | $\mathcal{L}_{dis}$ | Post-train $a_t$mAP (%) | Pre-train $a_t$mAP (%) |
|---|---|---|---|---|---|---|
| ❶ | - | - | - | - | 67.45 | 65.83 |
| ❷ | ✓ | - | ✓ | - | 68.43 | 18.27 |
| ❸ | - | - | - | ✓ | 67.78 | 66.92 |
| ❹ | - | ✓ | - | - | 64.89 | 61.19 |
| ❺ | - | ✓ | - | ✓ | **71.10** | **71.32** |

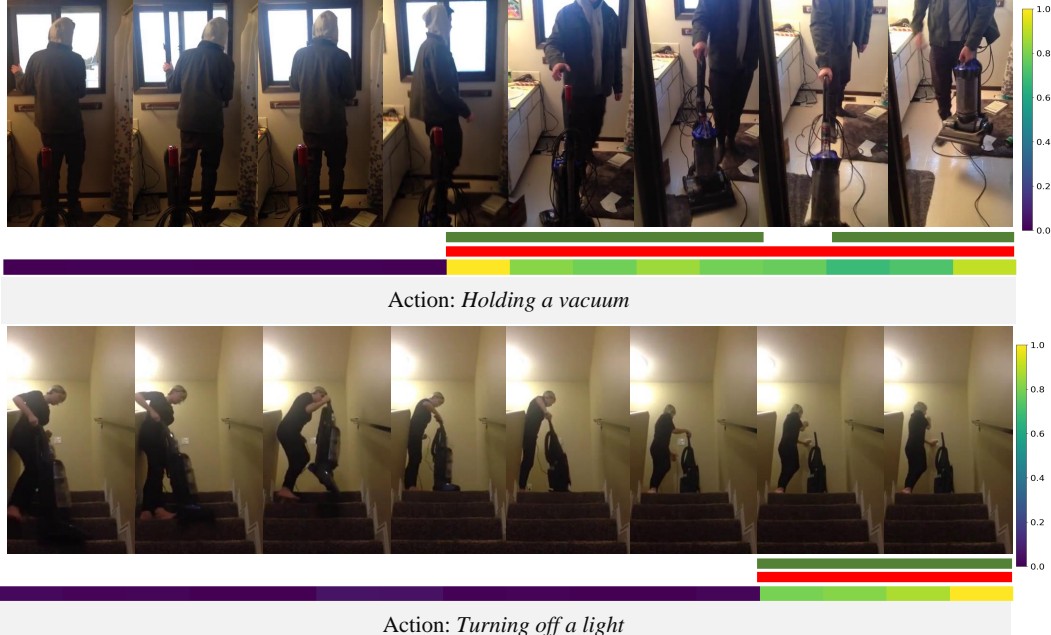

Action: *Holding a vacuum*

Action: *Turning off a light*

Figure 8: Frame-level weights $s$ predicted by the readout function for different non-distractor-injected SAPs within the same video. The heatmaps (*colored bars*) visualize the predicted weights, reflecting the relevance of each frame to the target action. The *red bars* indicate the ground-truth action segments, while the *green bars* denote our localization results obtained by thresholding $s$ at 0.7.

In Figure 10, we show a more complex example from a video with 15 labels, containing multiple concurrent interactions. In this example, the person interacts simultaneously with a table, chair, cup, book, and food. Using prompts, we disentangle the interaction between the person and the shoe from these overlapping interactions. Additionally, the subgraphs relevant to the specified action are successfully parsed from the full SSG. In Figure 11, we present an example from a video with 10 labels, which also contains multiple concurrent interactions. In this example, the person interacts simultaneously with a sofa, blanket, TV, and food. Using prompts, we disentangle the interaction between the person and the shoe from these overlapping interactions. Additionally, the subgraphs relevant to the specified action are successfully parsed from the full SSG.

# F Limitation

Our method operates solely on visual features and the derived Spatio-Temporal Scene Graphs (SSGs), without incorporating other modalities such as audio, text, or human pose. This reliance on a single modality may limit the flexibility and robustness of the model, especially in scenarios where multimodal cues are essential for disambiguating actions. In future work, we plan to extend our framework to integrate multimodal information, enabling more comprehensive and robust action disentanglement in complex environments.

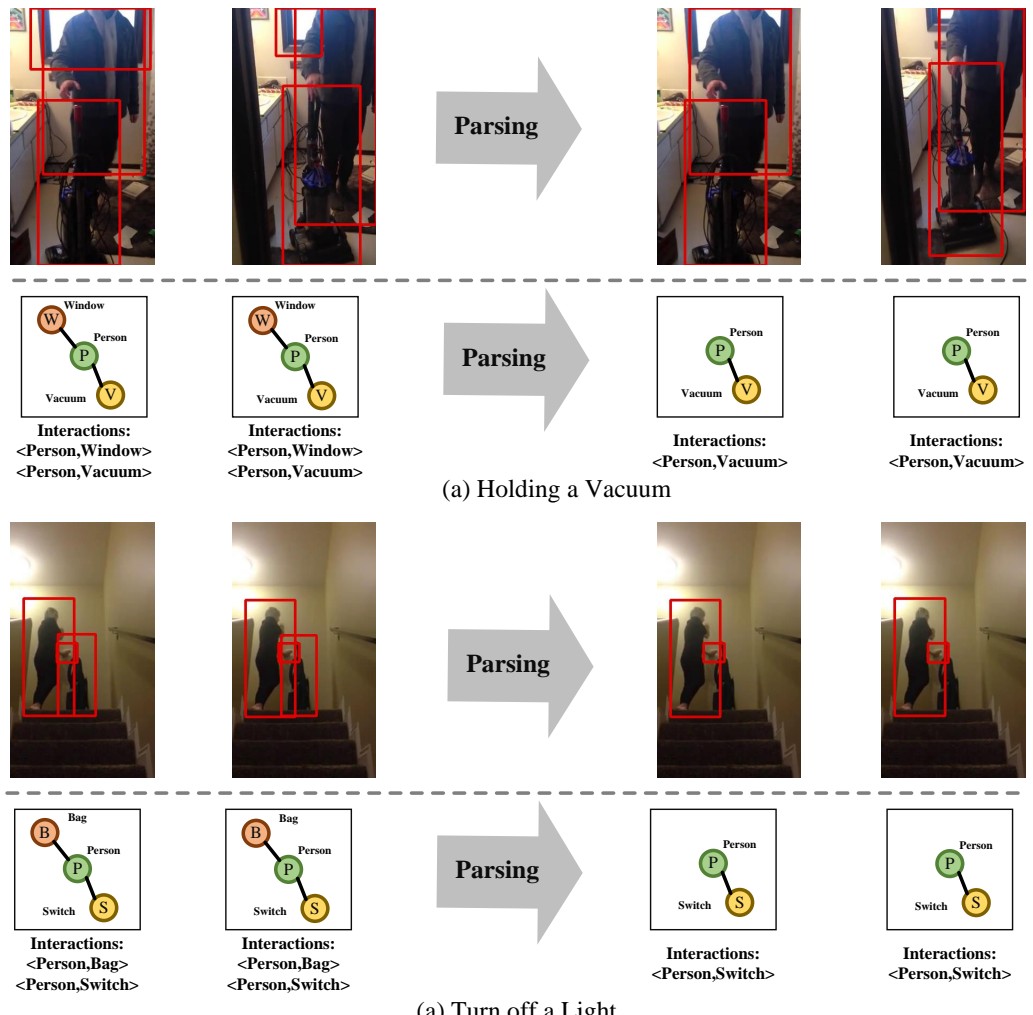

Figure 9: Key parsing steps for the two examples presented in Figure 8. For each case, the specified action is disentangled from multiple co-occurring interactions. The red boxes indicate the objects retained by VGPNN during the parsing process. Initially, the scene graph is fully connected with all nodes; after parsing, only a subset of relevant nodes remains. Below each image is the corresponding scene graph of that frame.

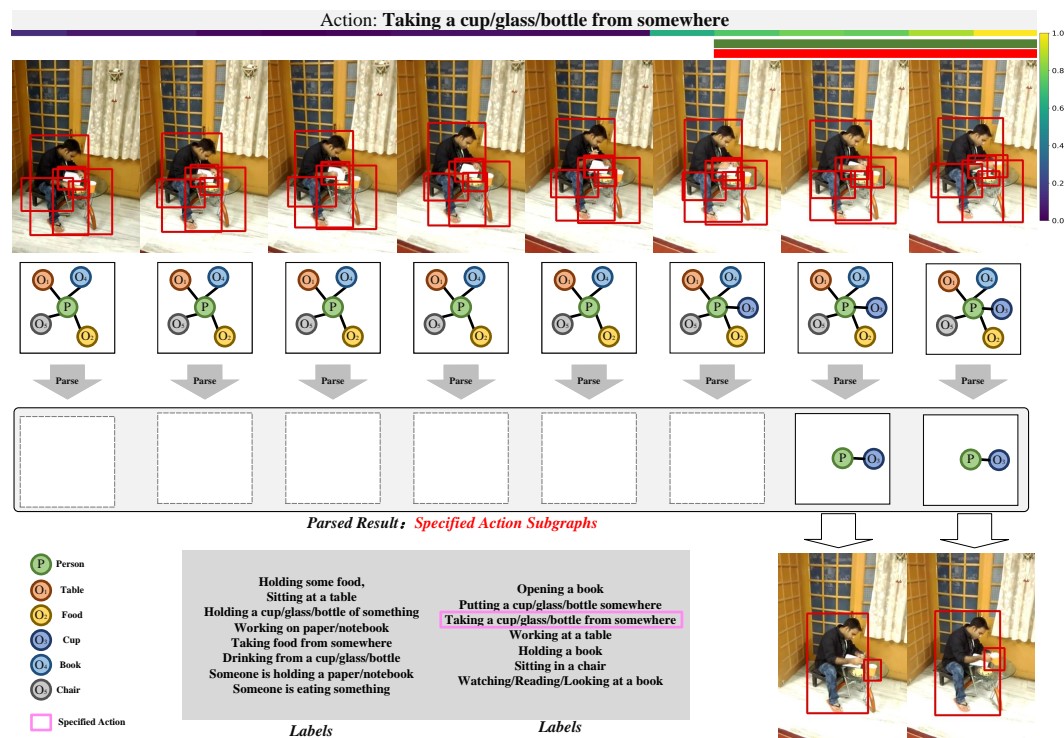

Figure 10: This is the parsing process of a 15 labels video. We disentangle a single interaction from a series of complex interactions in the scene by using prompts. The top part of the image shows our Temporal Action Localization results.

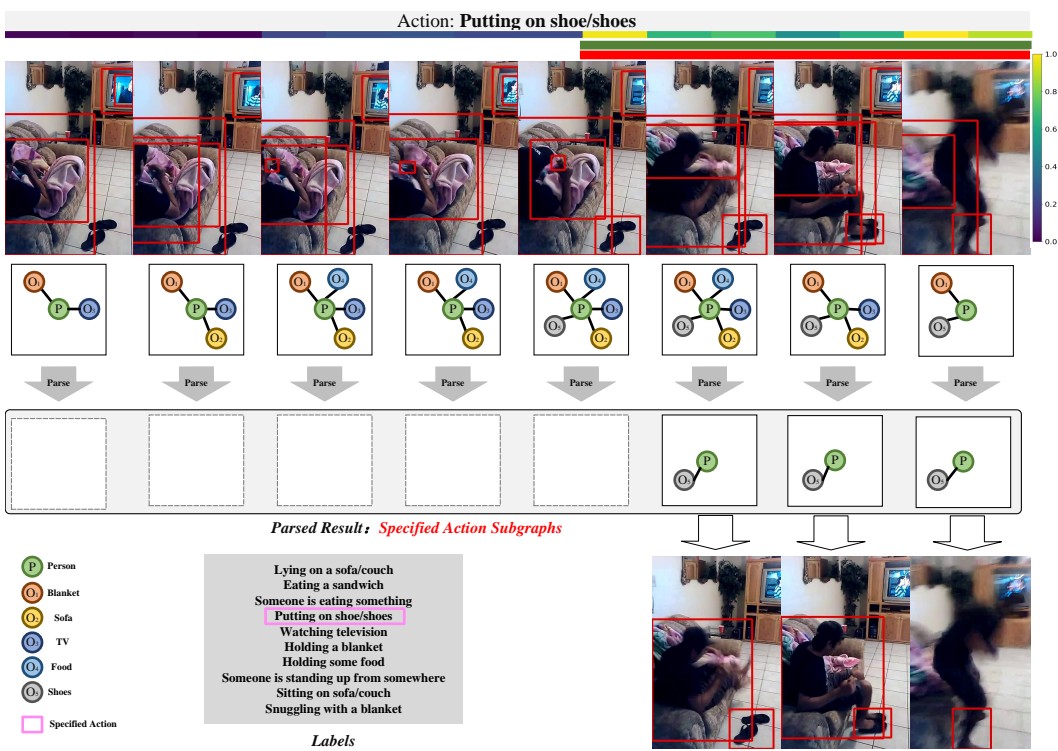

Figure 11: This is the parsing process of a 10 labels video. We disentangle a single interaction from a series of complex interactions in the scene by using prompts. The top part of the image shows our Temporal Action Localization results.

