# OpenReview forum: "Prompt-guided Disentangled Representation for Action Recognition"
_NeurIPS.cc/2025/Conference — NeurIPS 2025 poster_

### Official Review · Reviewer_SVW3 · 2025-06-13

**Clarity:** 3
**Significance:** 2
**Originality:** 3
**Rating:** 4
**Confidence:** 4

**Summary:**

This paper proposes a Prompt-guided Disentangled Representation framework for action recognition. Experimental results on video action recognition demonstrate the effectiveness of the proposed method, ProDA, compared to state-of-the-art approaches.

**Questions:**

•The contributions and novelty of the proposed method should be better refined and reorganized to highlight its significance.

• Existing multimodal large language models (MLLMs) have already demonstrated strong performance in action recognition, particularly in few-shot and zero-shot settings [1]. The authors should clearly state what advantages their method offers over such models.

•In the proposed Action Disentanglement module, it is unclear whether the Action Specification captures temporal relationships. Would changing the order of specified actions or introducing interaction between actions affect model performance?

•The idea of using prompts to augment video inputs and enhance model performance—i.e., prompt learning—has been widely explored in prior work. The authors fail to sufficiently discuss and cite relevant literature on this topic [3–5].

•To validate the generalizability of the proposed method, additional widely used datasets such as AVA [2] should be considered.

• The differences between VGNorm and previous GNN norm-based methods need to be clearly explained. If applicable, comparisons with more recent normalization techniques such as GRANOLA[6] should also be included.

Reference:
1. LLaVAction: evaluating and training multi-modal large language models for action recognition
2. Gu, Chunhui, et al. "Ava: A video dataset of spatio-temporally localized atomic visual actions." Proceedings of the IEEE conference on computer vision and pattern recognition. 2018.
3. Zhou, Kaiyang, et al. "Learning to prompt for vision-language models." International Journal of Computer Vision 130.9 (2022): 2337-2348.
4. Bahng, Hyojin, et al. "Exploring visual prompts for adapting large-scale models." arXiv preprint arXiv:2203.17274 (2022).
5. Khattak, Muhammad Uzair, et al. "Maple: Multi-modal prompt learning." Proceedings of the IEEE/CVF conference on computer vision and pattern recognition. 2023.
6. Eliasof, Moshe, et al. "Granola: Adaptive normalization for graph neural networks." arXiv preprint arXiv:2404.13344 (2024).

**Ethical Concerns:**

["NO or VERY MINOR ethics concerns only"]

**Final Justification:**

The authors have added experiments and addressed most of my concerns, and I raise my score.

**Limitations:**

The author did not discuss the limitations of the work.

**Paper Formatting Concerns:**

The author did not finish the paper checklist completely, with some Justifications remaining [TODO].

**Quality:**

2

**Strengths And Weaknesses:**

The method achieves the SOTA performance on multi-label action understanding with interpretability.
However, the novelty of the proposed method should be better refined and reorganized, and more comparison with existing methods and evaluation on other datasets should be classified, as shown in the questions below.
Moreover, there are some typo mistakes such as "State-of-the Art Methods" in line 298.

---

> ### Author Rebuttal · Authors · 2025-07-31
>
> We thank the reviewer for the positive comments on our method. The reviewer also raised concerns, which we address below.
>
> **Q1:** The contributions and novelty of the proposed method should be better refined and reorganized to highlight its significance.
>
> **A1:** In the final version, we will revise the introduction and contributions section to more clearly emphasize the novelty of our method. Our key innovations include: 1) a prompt-guided disentanglement framework for multi-action videos; 2) a dynamic prompt module that adaptively encodes specified actions into expressive prompts to guide disentanglement; and 3) a tailored normalization and loss design for structured and temporally consistent action representation.
>
> **Q2:** Existing MLLMs already perform well in action recognition, especially in few-shot and zero-shot cases. The authors should clearly explain what advantages their method has over these models (e.g., LLaVAction).
>
> **A2:** While LLaVAction have shown promising performance in few-shot and zero-shot action recognition, our approach differs from LLaVAction  in several key aspects, which we summarize from the following two perspectives:
> - **Motivation.** Our goal is to improve video understanding by disentangling actions from complex scenes. We focus on modeling human-object and human-human interactions, enabling more interpretable reasoning over interaction structures. In contrast, LLaVAction addresses the limited generalization of egocentric video datasets by converting class labels to natural language. They find that adding distractors during test harms performance on multi-choice QA tasks and propose method to mitigate this.
> - **Task Scope.** Our approach constructs SSGs and uses subgraph parsing to disentangle actions, yielding a framework that is naturally interpretable and extensible. For example, localization arises directly from attention weights during graph readout, without explicit supervision. In contrast, LLaVAction requires supervision for localization.
>
> Due to limited resources and time during the rebuttal period, we were unable to conduct a full evaluation using LLaVAction . **Instead, we carried out an experiment by replacing our multi-hot vector prompt with a CLIP-based text prompts.** Specifically, we converted all class labels into natural language and used CLIP to encode them as prompts. As shown in Table 1, our method achieves a significant performance advantage (+7.02 mAP). We attribute this to our method's ability to generate multiple prompts per action specification (AS) and dynamically aggregate them, resulting in stronger representational capacity.
>
> **Table 1:** Comparison between Multi-hot Vector Prompts and Text Prompts.
> |Prompt Method |mAP (%)|
> |-|-|
> |Multi-hot Vector Prompts| 71.10|
> |Text Prompts|64.08|
>
> **Q3:** In the proposed Action Disentanglement module, it is unclear whether the Action Specification captures temporal relationships. Would changing the order of specified actions or introducing interaction between actions affect model performance?
>
> **A3:**  Through **our frame-level localization results**, we observe that the model is able to **capture the temporal extent of actions**, indicating that the prompts effectively encode relationships among relevant atomic motions.
> Since the disentanglement prompts are represented as multi-hot vectors, changing the order of actions within the prompt does not affect the results.
>
> To verify that changing the order of specified actions or introducing interactions between actions does not affect model performance, we randomly shuffled the label meanings in the dataset. For example, the label "c000'" originally represented "Holding some clothes", after shuffle, "c000" represents "Sitting in a chair". We tested results under multiple random seed shuffles, the results are shown in Table 2. **We observe no significant difference in performance, indicating that our model is invariant to the order of specified actions, and that potential interactions between action specifications do not significantly impact the model's effectiveness.**
>
> **Table 2:** Comparison of model performance using original _v.s._ shuffled dataset label.
> |Label Type|mAP (%)|
> |-|-|
> |Original|71.10|
> |Shuffled|71.50|
>
> **Q4:** The idea of using prompts to augment video inputs and enhance model performance—i.e., prompt learning—has been widely explored in prior work. The authors fail to sufficiently discuss and cite relevant literature on this topic [3–5].
>
> **A4:** While prompt learning is widely studied, prior work mainly adapts pretrained models for zero- or few-shot tasks using language prompts. In contrast, we use structured multi-hot prompts to explicitly guide action disentanglement in multi-label recognition. Our model is trained from scratch and does not rely on any pretrained language or vision models. We will clarify this distinction in the related work section.
>
> **We conducted additional experiments using a CLIP-based text encoder**. Specifically, we converted all action labels into natural language form and encoded them with CLIP to replace the original multi-hot vector prompt representation.  As shown in **Table 1 above**, our proposed method significantly outperforms the CLIP-based variant.
>
> **Q5:** To validate the generalizability of the proposed method, additional widely used datasets such as AVA  should be considered.
>
> **A5:** We conduct our experiments on SportsHHI instead of AVA. While AVA mainly contains low-density interactions in movie scenes, **SportsHHI captures dense, real-world Human-Human Interactions (HHI) from basketball and volleyball games, making it more challenging.** Thus, SportsHHI serves as a better benchmark for evaluating our method’s effectiveness in complex interaction scenarios.
>
> Based on the baseline, we incorporated our Dynamic Prompt Module (DPM) and Action Disentanglement Loss (AD Loss). To ensure a fair comparison, we conducted an additional experiment by increasing the parameter size of the baseline model to match that of our modified version. The detailed results are shown in Table 3. **Our method consistently outperforms the baseline across all metrics, demonstrating the generalization ability of our approach. Even under comparable parameter settings, our method still achieves significant performance improvements.** This indicates that the performance gain primarily stems from our disentanglement mechanism, rather than merely an increase in model capacity.
>
> **Table 3:** Human-Human Interaction Classification results in SportsHHI.
> SportsHHI\* is our reimplementation based on the official SportsHHI code, with minor modifications (e.g., parameter size) to match the capacity of ProDA for fair comparison.
> |Method|mAP|R@150|R@100|R@50|R@20|Params|Time(s)|
> |-|-|-|-|-|-|-|-|
> |STTran| 3.31| 71.29|62.91|42.67|22.14|-|-|
> |HORT|3.75|78.57|67.78|50.33|26.96|-|-|
> |SlowFast|5.00| 81.66|74.00|52.74|26.82|-|-|
> |ACARN|5.44|82.85|75.22|56.53|31.77|-|-|
> |**SportsHHI**(baseline)|10.69|89.25|82.93|68.13|43.72|111.18M|0.0478|
> |**SportsHHI\***(reimplemented)|10.16 (↓0.53)| 90.07 (↑0.82) |84.83 (↑1.9)|68.40(↑0.37)|40.16(↓3.56)|123.11M|0.0484|
> |**Ours**(baseline+ProDA)|**13.55 (↑2.86)**|**92.54 (↑3.29)**|**87.91 (↑4.98)**|**72.69(↑4.56)**|**47.81(↑4.09)**|**123.19M**|**0.0520**|
>
> **Q6:** The differences between VGNorm and previous GNN norm-based methods need to be clearly explained. If applicable, comparisons with more recent normalization techniques such as GRANOLA should also be included.
>
> **A6:** We compared widely used normalization techniques in graph learning, including BatchNorm, LayerNorm, and GraphNorm, specifically in the context of SSG (As shown in Table 4).
> - During training, BatchNorm computes statistics over the mini-batch, which can suppress sparse yet informative frame-level features in temporal data. To address this, we introduce learnable scaling for the mean and compute variance in real-time on each input to better preserve salient temporal signals.
> - LayerNorm operates at the node level, normalizing each node independently, whereas our approach performs normalization globally over the entire SSG layer.
> - GraphNorm computes the mean in real-time and applies learnable parameters to preserve salient features per graph. However, it lacks normalization across the temporal batch dimension. To address this, we use the accumulated mean during training, enabling temporal batch-level normalization.
>
> **Table 4:** Differences among LayerNorm, BatchNorm, GraphNorm, and VGNorm.
> "Computation of Mean" and "Computation of Std" refer to how the mean and standard deviation are computed. "In-Batch" indicates that the statistics are calculated from the current input batch, while "In-Training" uses accumulated statistics during training. * indicates that a learnable scaling parameter mean_scale is applied to the mean during normalization.
> ||LayerNorm|BatchNorm|GraphNorm|VGNorm|
> |-|-|-|-|-|
> |Norm Aspect|Node|Batch|Graph|Batch & Graph|
> |Computation of Mean|In-Batch|In-Training|In-Batch*|In-Training*|
> |Computation of Std|In-Batch|In-Training|In-Batch|In-Batch|
>
> As the official code for GRANOLA is currently unavailable (the GitHub repository linked in the paper is empty), we implemented our own version for evaluation. The results are shown in Table 5.**Our method achieves better results**. Compared to GraphNorm, GRANOLA also performs normalization at the graph level. GRANOLA leverages Random Node Features (RNF) to adaptively capture the structural properties of input graphs, and generates dynamic affine transformation parameters through GNN and MLP modules, thereby enhancing its expressive power. However, **GRANOLA ignore normalization across the temporal dimension**, which limits their effectiveness in video-based settings.
>
> **Table 5:** Comparison between textual encoding and multi-hot encoding.
> |Norm Type|mAP (%)|
> |-|-|
> |VGNorm|71.10|
> |GRANOLA| 70.29|
> |GraphNorm|68.61|

---

> > ### Author Response · Authors · 2025-08-03
> >
> > Thank you very much for your review and helpful comments. We have done our best to address your concerns in the rebuttal and would greatly appreciate it if you could let us know whether any questions remain or if any part of our response could be improved. We value your feedback and thank you again for your time.

---

> > > ### Comment · Area_Chair_yumv · 2025-08-05
> > > **Please reply to rebuttal**
> > >
> > > Dear Reviewer,
> > >
> > > please reply to the rebuttal of the authors. Please let them know as soon as possible if any concerns remain or if there is anything that needs further clarification.
> > >
> > > Thanks
> > >
> > > AC

---

### Official Review · Reviewer_Lz46 · 2025-06-28

**Clarity:** 4
**Significance:** 2
**Originality:** 4
**Rating:** 5
**Confidence:** 4

**Summary:**

This paper introduces ProDA, a framework that disentangles specified actions from multi-action videos using prompt signals and spatio-temporal scene graphs. The design is well structured, with components such as the Dynamic Prompt Module and a graph-based encoder that together achieve strong performance on standard benchmarks. The overall motivation is sound, and the system is practically useful for structured video understanding.

However, the core idea of using prompts is only superficially explored. The prompts are implemented as simple multi-hot vectors without any semantic grounding, and the method does not incorporate language models that could reason about action semantics and generate richer prompt representations. This limits the conceptual depth of the work. While the prompt-guided framing is promising, its realization remains shallow. A more complete integration of language understanding could have made the contribution substantially stronger.

Overall, this is a well-executed paper with a good idea at its core, but it lacks depth in exploring the full potential of prompt-based reasoning.

**Questions:**

1. The key concern is encoding prompts with fixed multi-hot vectors. This choice appears to block open-set recognition and prevents the framework from exploiting prior knowledge available in large language models, thereby limiting the broader significance of the contribution. Nevertheless, the work presents a well-executed and novel perspective on action recognition. If additional analysis is provided, such as evaluating the potential of language-model-guided prompts or demonstrating the flexibility of the current encoding scheme, I would be happy to raise my rating.

2. Another concern relates to the fusion strategy used in Equation 1, where visual features and prompt features are combined through simple addition. This choice may limit the expressiveness of the interaction between action specifications and scene graph features. It would be valuable to explore alternative fusion methods such as concatenation followed by a linear layer, or cross-attention mechanisms. A comparison of these approaches could provide insight into whether the current additive design is optimal or if stronger interaction modeling would further enhance performance.

3. VGNorm computes temporal statistics using mean-based aggregation to enforce consistency across frames. While this design is standard in normalization, it remains unclear whether mean alone is the most effective choice for modeling temporal dynamics in spatio-temporal scene graphs. Mean aggregation may dilute salient or bursty features that occur sparsely over time. It is not clear whether combining mean with other statistics such as max, or using a learnable aggregation function, would offer any advantage in this context. Could the authors clarify whether such alternatives were considered, and whether there is theoretical or empirical justification for relying solely on mean-based aggregation?

4. The proposed method relies on object bounding boxes to construct spatio-temporal scene graphs, which serve as the foundation for downstream disentanglement. However, the quality of these bounding boxes during training—including their number, category diversity, and spatial distribution—is likely to have a significant impact on the effectiveness of feature disentanglement. The paper does not provide any analysis on how variations in bounding box quality influence model performance. Has this factor been studied, and is there any quantitative or qualitative assessment of how sensitive the method is to the quality of the input object annotations?

**Ethical Concerns:**

["NO or VERY MINOR ethics concerns only"]

**Final Justification:**

The authors have addressed all of my concerns, and the paper is solid overall. Therefore, I am raising my rating to accept.

**Limitations:**

Yes

**Quality:**

3

**Strengths And Weaknesses:**

Strengths:
1. The paper proposes a disentangled representation learning framework (ProDA) for multi-label action recognition, with the goal of isolating specific actions from complex scenes. The motivation is clear and addresses a practical limitation of prior methods that process all actions jointly.

2. The framework is well structured, integrating modules such as the Dynamic Prompt Module (DPM), Video Graph Parsing Neural Network (VGPNN), and a novel normalization strategy (VGNorm) tailored to spatio-temporal scene graphs.

3. The empirical results are strong, with consistent improvements over existing methods across multiple benchmarks. The ablation studies are thorough and support the utility of each component.

Weaknesses:
1. Although the paper frames the method as “prompt-guided,” the prompts are implemented using simple multi-hot vectors that lack semantic richness. This shallow encoding limits the model’s ability to reason about unseen or open-set actions, as it cannot generalize beyond the fixed label space.

2. The method does not incorporate any language model or text encoder to provide semantic grounding for action prompts. As a result, the full potential of prompt-based guidance is underutilized, and the framework misses the opportunity to leverage external knowledge or compositional semantics.

---

> ### Author Rebuttal · Authors · 2025-07-31
>
> We thank the reviewer for highlighting the strengths of our work. They have nonetheless asked some questions, which we address below.
>
> **Q1:** The key concern is encoding prompts with fixed multi-hot vectors. This choice appears to block open-set recognition and prevents the framework from exploiting prior knowledge available in large language models, thereby limiting the broader significance of the contribution. Nevertheless, the work presents a well-executed and novel perspective on action recognition. If additional analysis is provided, such as evaluating the potential of language-model-guided prompts or demonstrating the flexibility of the current encoding scheme, I would be happy to raise my rating.
>
> **A1:** We appreciate the reviewer for raising this point. In response, we conducted an additional experiment using textual prompts. We converted all Action Specification (AS) into text form and then used the CLIP (ViT-B/16)  text encoder (as used in our main method) to encode the textual HOI action name corresponding to each task. To adapt the textual prompts to our task, we appended a CLIP Adapter after the text prompt embeddings. Finally, we used an addition-based fusion, where the textual prompt was directly added to the visual features. Given an $AS  = \\{as_i\\}_{i=1}^K$, where each $as_i$ represents textual Human-Object Interaction (HOI) action name (e.g., "Holding some clothes", "Putting clothes somewhere"), the calculation process is as follows.
>
> \begin{equation}
> 	\text{text prompt}=\mathtt{Adapter}(\sum_{i=0}^{K}\mathtt{CLIP}(as_i))
> \end{equation}
>
> The detailed results of this experiment are shown in Table 1. The results show that the multi-hot vector-based AS achieves a considerable advantage. This may be attributed to two factors:  1)the Action Specification (AS) includes many distractor items, making it difficult for textual-based AS to generalize effectively; 2) our multi-hot vector prompts generates T candidate prompts for each action category (we use $T=10$ in our method), which **provides greater representational capacity than a single prompt embedding. Moreover, we dynamically aggregate the candidate prompts based on both the input features and AS weights, enabling a more adaptive and informative representation**. Despite lacking the prior knowledge of large-scale language models, this approach still achieves strong performance. In future work, we plan to explore integrating large vision-language models into our framework to combine structural disentanglement with semantic generalization.
>
> **Table 1:** Comparison between textual encoding and multi-hot encoding.
>
> |Prompt Method |mAP (%)|
> |-|-|
> |Multi-hot Vector Prompts| 71.10|
> |Text Prompts|64.08|
>
> **Q2:** Another concern relates to the fusion strategy used in Equation 1, where visual features and prompt features are combined through simple addition. This choice may limit the expressiveness of the interaction between action specifications and scene graph features. It would be valuable to explore alternative fusion methods such as concatenation followed by a linear layer, or cross-attention mechanisms. A comparison of these approaches could provide insight into whether the current additive design is optimal or if stronger interaction modeling would further enhance performance.
>
> **A2:** Under the same training setups and model architecture, we additionally tested two alternative fusion methods (Results are shown in Table 2). The first method (line 2, Concate&Linear) concatenates the prompt and the original features, followed by a single linear layer with GELU activation to fuse them. The second method (line 3, Cross-Attention) employs a single-layer cross-attention module, where the prompt acts as the query (Q), and the original features serve as the key (K) and value (V). The results are shown in Table 2. We observe that both concatenation and addition yield similar performance, both outperforming the cross-attention approach. This may be due to the cross-attention introducing significantly more parameters, which increases training difficulty and the risk of overfitting. Comparatively, the addition method uses the fewest parameters and achieves the best results.
>
> **Table 2:** Results of different fusion methods in the DPM
> |Fusion Method|mAP (%)|
> |-|-|
> |addition| 71.10|
> |Concate & Linear|71.00|
> |Cross-Attention|70.04|
>
> **Q3:** VGNorm computes temporal statistics using mean-based aggregation to enforce consistency across frames. While this design is standard in normalization, it remains unclear whether mean alone is the most effective choice for modeling temporal dynamics in spatio-temporal scene graphs. Mean aggregation may dilute salient or bursty features that occur sparsely over time. It is not clear whether combining mean with other statistics such as max, or using a learnable aggregation function, would offer any advantage in this context. Could the authors clarify whether such alternatives were considered, and whether there is theoretical or empirical justification for relying solely on mean-based aggregation?
>
> **A3:** We additionally evaluated different aggregation strategies within the normalization component under the same training setups and model architecture. The first variant augments the standard mean aggregation with per-frame maximum values, i.e., computing the maximum within each frame and combining it with the mean. The second variant removes global averaging entirely and instead uses a learnable aggregation function, implemented as a one-layer MLP that predicts fusion weights to combine frame-level features.
>
> As shown in Table 3 both Mean (line 1) and Mean&Max (line 2) aggregation yield similar performance, and outperform the MLP-based method (line 3). This may be due to the fact that the learnable MLP fails to maintain temporal consistency across frames. **While mean aggregation may dilute salient or bursty features that occur sparsely in time, we mitigate this issue by introducing learnable scaling parameters on the mean, which helps it match the performance of the mean+max approach.**
>
> Considering both accuracy and computational efficiency, we adopt simple mean aggregation as the final choice in our framework.
>
> **Table 3:** Comparison of Aggregation Strategies in VGNorm
> |Methods|mAP (%)|
> |-|-|
> |Mean|71.10
> |Mean&Max|70.80|
> |MLP|67.18|
>
> **Q4:** The proposed method relies on object bounding boxes to construct spatio-temporal scene graphs, which serve as the foundation for downstream disentanglement. However, the quality of these bounding boxes during training—including their number, category diversity, and spatial distribution—is likely to have a significant impact on the effectiveness of feature disentanglement. The paper does not provide any analysis on how variations in bounding box quality influence model performance. Has this factor been studied, and is there any quantitative or qualitative assessment of how sensitive the method is to the quality of the input object annotations?
>
> **A4:** Thank you for the insightful question regarding the sensitivity of ProDA to errors in SSG construction. **We follow the same approach as LaIAR** by employing Faster R-CNN to detect and classify objects in videos, and construct SSGs with inherent errors. The detailed results are shown in Table 4. As can be seen, **our method achieves the best performance even when using noisy SSGs. Moreover, our performance drop is the smallest among all compared methods**. We attribute this to our action disentanglement mechanism, which explicitly separates actions from the complex scene context. This allows our model to understand actions in a simpler and less error-prone representation. In contrast, feature-based methods attempt to interpret actions directly from the noisier and more entangled SSGs.
>
> **Table 4:** Results of SSGs under different conditions.
> GT SSGs are ground-truth SSGs without detection errors. FRCN SSGs are obtained using a detector (i.e., Faster R-CNN) and may contain errors.
> |Method|FRCNN SSGs mAP (%)|GT SSGs mAP (%)|
> |-|-|-|
> |LaIAR+NL|45.1|67.4 (+22.3)|
> |OR2G+NL|44.9|67.5 (+22.6)|
> |Ours|50.2| 71.1 (+20.9)|

---

> > ### Comment · Reviewer_Lz46 · 2025-08-03
> >
> > Thank you to the authors for the detailed response. I appreciate the added discussion on open-set encoding of text prompts, which directly addresses my primary concern. I will raise my rating accordingly and hope the paper will be accepted.

---

> > > ### Author Response · Authors · 2025-08-03
> > >
> > > Thank you very much for your thoughtful follow-up and for taking the time to reconsider your evaluation. We’re truly grateful that our clarification addressed your concern, and we sincerely appreciate your support and updated rating.

---

### Official Review · Reviewer_kb31 · 2025-07-04

**Clarity:** 2
**Significance:** 2
**Originality:** 3
**Rating:** 4
**Confidence:** 3

**Summary:**

The paper presents a framework for disentangling specified actions from multi-action video scenes using prompt-guided spatio-temporal scene graphs. By leveraging prompts to focus on particular actions, the method isolates relevant object and interaction representations, making it easier to model complex scene dynamics. This targeted disentanglement leads to improved performance on standard multi-label action recognition benchmarks.

**Questions:**

Already mentioned in weaknesses

**Ethical Concerns:**

["NO or VERY MINOR ethics concerns only"]

**Final Justification:**

This paper presents an interesting new angle on the role of disentangled representations for the task of action recognition and I beleive the community would find this interesting.

**Limitations:**

Yes

**Quality:**

3

**Strengths And Weaknesses:**

Strengths :

1. The paper is well written, and the motivation, key ideas, and contributions are clearly articulated and easy to follow.

2. The idea of disentangling desired actions by leveraging prompts is very compelling. Based on the results provided, the approach appears to be effective. While vanilla methods may struggle in ambiguous scenarios, the proposed DPM framework seems to overcome these limitations by generating adaptive prompts conditioned on both the action specification and the video content.

3. The ablation study is thorough and clearly isolates the contributions of each module, helping to attribute the performance gains to specific components of the framework.

Weaknesses :

1.  While the method shows some improvements in mAP, the added complexity of the proposed approach may not be fully warranted for the action recognition task, particularly in the absence of a detailed comparison of model efficiency or computational cost. In other words, can some of the performance gains be coming from increased model capacity rather than the disentaglement loss?

2. Since the central claim of the paper is that explicit disentanglement of actions improves performance, a more compelling demonstration would involve applying the disentanglement loss to an existing baseline model without increasing the parameter count. This would better isolate the effect of the disentanglement itself. As it stands, the comparison in Table 3 involves different architectures and training setups, making it unclear whether the performance gains are due to the disentanglement mechanism or other architectural differences and whether the disentanglement loss is always strictly necessary for all models.

3. The evaluation is limited to a single dataset, and the performance gains over certain baselines are relatively modest. Including results on one or two additional datasets would strengthen the generalizability claims and further validate the robustness of the approach.

4. Although the core ideas are well-motivated, the transition between some concepts could be smoother. Improving the overall narrative flow would help the reader better follow the progression of the proposed framework.

---

> ### Author Rebuttal · Authors · 2025-07-31
>
> We thank the reviewer for their feedback, and we are glad to see they have recognized the significance of our contribution. They have nonetheless asked some questions, which we address below.
>
> **W1:** Without detailed comparisons of model efficiency or computational cost, it remains unclear whether some of the performance gains arise from increased model capacity rather than the Action Disentanglement Loss?
>
> **A1:** We thank the reviewer for the insightful comment. To demonstrate that the performance gain of our method primarily comes from the proposed action disentanglement mechanism, rather than merely increased model capacity, we conduct additional experiments on SportsHHI , a recent benchmark focusing on human-human interaction in sports videos. It provides fine-grained annotations of interactions between players, SportsHHI contains 34 high-level interaction classes from basketball and volleyball sports. 118,075 human bounding boxes and 50,649 interaction instances are annotated on 11,398 keyframes. Compared to conventional action datasets, SportsHHI offers richer relational labels and poses greater challenges due to its dense, multi-person interactions.
>
> We conduct two experiments for fair and thorough evaluation:
> - Module Integration: We extend the official SportsHHI baseline  (implemented with MMAction2 ) by integrating our proposed components, including two Dynamic Prompt Modules (DPM)—one for the Unspecified Action Prompt (UAP) and one for the Specified Action Prompt (SAP)—as well as the Action Disentanglement Loss (AD Loss).
> - Parameter-Controlled Baseline: To rule out the impact of model capacity, we construct a parameter-matched variant of the baseline by increasing its model size to match that of our approach, without incorporating any of our proposed modules.
>
> We address the reviewer’s concern from the following two perspectives.
>
> First, as results shown in Table 1, we increased the parameter count by 10.8\% (with the DPM contributing only 1.46M parameters, which is 1.18\% of the total), yet achieved a significant accuracy improvement of +2.86 mAP. **Under this comparable parameter setting, our method significantly outperforms the baseline (+3.39mAP).** This highlights the generalization ability of our approach and further confirms that the performance gain mainly stems from the proposed AD Loss, rather than simply from increased model capacity.
>
> Second, as shown in Table 2 (Evaluation on Charades Dataset), without adding the Action Disentanglement Loss (AD Loss), our method already achieves a competitive accuracy of 67.45 mAP, comparable to the previous state-of-the-art (SOTA) result of 67.5 mAP. Furthermore, in our ablation study under the same architecture and training settings, **incorporating the AD Loss improves the performance by +3.65 mAP (67.45 _v.s_. 71.10).** This demonstrates that the performance gain is largely attributed to the effectiveness of our disentanglement loss beyond the base model capability.
>
> **Table 1:**  Human-Human Interaction Classification results in SportsHHI. SportsHHI\* is our reimplementation based on the official SportsHHI code, with minor modifications (e.g., parameter size) to match the capacity of ProDA for fair comparison.
> |Method|mAP|R@150|R@100|R@50|R@20|Params|Time(s)|
> |-|-|-|-|-|-|-|-|
> |STTran| 3.31| 71.29|62.91|42.67|22.14|-|-|
> |HORT|3.75|78.57|67.78|50.33|26.96|-|-|
> |SlowFast|5.00| 81.66|74.00|52.74|26.82|-|-|
> |ACARN|5.44|82.85|75.22|56.53|31.77|-|-|
> |**SportsHHI**(baseline)|10.69|89.25|82.93|68.13|43.72|111.18M|0.0478|
> |**SportsHHI\***(reimplemented)|10.16 (↓0.53)| 90.07 (↑0.82) |84.83 (↑1.9)|68.40(↑0.37)|40.16(↓3.56)|123.11M|0.0484|
> |**Ours**(baseline+ProDA)|**13.55 (↑2.86)**|**92.54 (↑3.29)**|**87.91 (↑4.98)**|**72.69(↑4.56)**|**47.81(↑4.09)**|**123.19M**|**0.0520**|
>
>
> **Table 2:** Impact of AD Loss on the Performance of the Charades Dataset
> |Method|mAP (%)|
> |-|-|
> |OR2G+NL|67.5|
> |Ours w/o AD Loss|67.45|
> |Ours w/ AD Loss|71.10|
>
> **W2:** To better isolate the effect of disentanglement itself, it would be more appropriate to add the AD Loss to a baseline without increasing model parameters. As shown in Table 3 of the original manuscript, it is unclear whether the performance gains come from architectural changes or from the disentanglement mechanism. Furthermore, it remains uncertain whether the disentanglement loss is strictly necessary for all model variants.
>
> **A2:** We appreciate the reviewer’s valuable feedback. We apologize the Table 3 of the original manuscript did not clearly convey the intended comparison, and we clarify that settings ①-⑤ all use the same backbone and training configuration. We will explicitly clarify this in the final version of the paper, if it is accepted. **We conducted experiments on the SportsHHI dataset to verify the effectiveness of applying the disentanglement loss to an existing baseline model without increasing the number of parameters.**
>
> To address the reviewer’s questions, we respond in two parts:
> - In the original ablation on the Charades dataset, adding the AD Loss improves mAP by +3.65 over the same model without it (As shown in **Table 2 above**). Additionally, we conducted new experiments on the SportsHHI dataset, where we added our proposed modules to the baseline. To control for parameter count, we also modified the baseline by expanding the FFN hidden dimension in the Transformer to match the parameter size of our method (As shown in **Table 1 above**). **The results show that our method still outperforms the parameter-matched baseline (+3.39mAP), suggesting that the performance gains stem not just from increased capacity, but from the effectiveness of our design.**
> - Regarding the AD Loss, it consists of two parts: a reconstruction loss and a disentanglement loss. Our goal is to explicitly disentangle action features, providing more concise and focused representations for specified actions, thus improving multi-label action understanding. **The reconstruction loss encourages the model to retain necessary information during separation. However, if only reconstruction is used, the model tends to preserve redundant features, undermining the purpose of action disentanglement.** Therefore, we introduce a disentanglement loss to minimize correlations between different action branches, ensuring that shared or redundant information is reduced.
>
> **W3:** The evaluation is limited to a single dataset, and the performance gains over certain baselines are relatively modest. Including results on one or two additional datasets would strengthen the generalizability claims and further validate the robustness of the approach.
>
> **A3:** We thank the reviewer for pointing this out. To address the reviewer’s concern and demonstrate the generalization ability of our approach,
>
> In the original submission, we evaluated our method on Charades to focus on Multi-label classification.
> To address the reviewer’s concern and demonstrate the generalization ability of our approach, we conduct additional experiments on another dataset, SportsHHI . In the original manuscript, we evaluated our method on Charades (Action Genome), which primarily focuses on human-object interactions. To assess the effectiveness of our approach in a different context, we choose SportsHHI, a dataset that emphasizes **human-human interactions**. Notably, SportsHHI is more complex and challenging, **as it centers on dense, multi-person interactions annotated in real-world sports videos.** In contrast, many conventional datasets (e.g., AVA ) contain fewer or relatively simpler human-human interactions.
>
> As shown in **Table 1 above**, **Our method achieves a +2.86 mAP improvement with only a 10.80% increase in total parameters.** Notably, the introduced DPM modules contribute only 1.46M parameters—merely 1.18% of the overall model size—highlighting the lightweight nature of our additions. Furthermore, **compared to a parameter-matched baseline with a similar model size, our approach still demonstrates a clear advantage (+3.39 mAP)**. These results indicate that the performance improvement mainly stems from better action disentanglement rather than an increase in model capacity.
>
> **W4:** Although the core ideas are well-motivated, the transition between some concepts could be smoother. Improving the overall narrative flow would help the reader better follow the progression of the proposed framework.
>
> **A4:** We thank the reviewer for the constructive feedback and for recognizing the motivation behind our core ideas. We agree that improving the narrative flow and smoothing transitions between concepts can significantly enhance readability. We will carefully revise the manuscript to better organize the presentation and improve the transitions between key sections in the final version, if it is accepted.
> These changes will help make the progression of our proposed framework clearer and more intuitive, facilitating a better understanding for readers.
>
> Our motivation is to disentangle specified actions from complex video scenes in order to better understand individual actions. To achieve this, we divide each video into two parts using a Specified Action Prompt (SAP) and an Unspecified Action Prompt (UAP), former is used to guide the model in disentangling the specified actions, while the latter disentangles all remaining actions. We use a Dynamic Prompt Module (DPM) to transform SAP and UAP into feature vector, which are then directly added to the video features to guide the model in disentanglement. To ensure that the model performs disentanglement rather than feature extraction, during training we enforce reconstruction of the original, non-disentangled features from the separated ones. Furthermore, to prevent excessive redundant information from hindering effective separation, we introduce a Disentanglement Loss that encourages more thorough disentanglement. For clarity, we include examples of how UAP and SAP are generated in the supplementary materials.

---

> > ### Author Response · Authors · 2025-08-03
> >
> > Thank you again for your review and constructive feedback. We’ve carefully addressed your comments in the rebuttal, and we would be grateful if you could let us know if any concerns remain or if there is anything further we could clarify or improve. We truly appreciate your time and consideration.

---

> > > ### Comment · Area_Chair_yumv · 2025-08-05
> > > **Please reply to rebuttal**
> > >
> > > Dear Reviewer,
> > >
> > > please reply to the rebuttal of the authors. Please let them know as soon as possible if any concerns remain or if there is anything that needs further clarification.
> > >
> > > Thanks
> > >
> > > AC

---

> > ### Comment · Reviewer_kb31 · 2025-08-07
> >
> > I thank the authors for elaborating on my concerns and addressing them satisfactorily. I updated my score accordingly.

---

> > > ### Author Response · Authors · 2025-08-07
> > >
> > > Thank you very much for taking the time to reconsider our response and for updating your score. We sincerely appreciate your thoughtful feedback and constructive comments throughout the review process. We're glad that our clarifications addressed your concerns.

---

### Official Review · Reviewer_MhYr · 2025-07-09

**Clarity:** 3
**Significance:** 3
**Originality:** 3
**Rating:** 4
**Confidence:** 5

**Summary:**

This paper addresses the challenge of accurately recognizing specific actions in complex, multi-action video scenes, where conventional methods typically extract unified features and struggle to capture action-specific semantics and inter-object interactions. To tackle this, the authors propose a novel framework called ProDA (Prompt-guided Disentangled Representation for Action Recognition), which enables the model to disentangle and analyze any specified action within a scene by leveraging Spatio-temporal Scene Graphs (SSGs) and prompt-based conditioning. Specifically, ProDA introduces a Dynamic Prompt Module (DPM) to generate action-aware prompts, a customized Video Graph Parsing Neural Network (VGPNN) to extract action-relevant subgraphs, and an Action Disentanglement Loss that guides the network to learn semantically meaningful and reconstructable representations. Without requiring explicit action localization labels, ProDA achieves state-of-the-art performance on multi-label action recognition benchmarks while maintaining strong interpretability and localization capabilities.

**Questions:**

1. Can the authors provide inference speed or computational cost analysis?
2. Could the authors include a qualitative visualization (e.g., t-SNE) comparing disentangled vs. unified features?
3. Why is the evaluation limited to a single dataset?
4. How sensitive is ProDA to errors in SSG construction?

If these issues are properly addressed, I would be happy to raise my evaluation score.

**Ethical Concerns:**

["NO or VERY MINOR ethics concerns only"]

**Final Justification:**

The authors have added experiments and addressed some of my concerns, but there is still room for improvement in terms of experiment completeness and visualization (I understand that it may not be convenient to add images during the rebuttal period). I maintain my score, but this paper can indeed be accepted.

**Limitations:**

Yes

**Paper Formatting Concerns:**

No.

**Quality:**

3

**Strengths And Weaknesses:**

Strengths:
1. The paper is motivated by a clear and relevant problem — the difficulty of recognizing specific actions in multi-action scenes using unified features.
2. The writing is well-structured and easy to follow.
3. The proposed method does not require action localization supervision.

Weaknesses:
1. The paper lacks analysis of the model's inference efficiency, which is important given the complexity of the proposed pipeline.
2. While the motivation for action-centric feature disentanglement is sound, the paper does not provide visual evidence (e.g., t-SNE) to demonstrate the effectiveness of the disentangled representations compared to conventional unified features.
3. Evaluation is limited to a single dataset, raising concerns about generalizability.
4. Additionally, the potential impact of SSG quality on model performance is not analyzed or discussed, which could affect robustness in real-world applications.

---

> ### Author Rebuttal · Authors · 2025-07-31
>
> We thank the reviewer for the positive comments on our motivation, writing, and the design of our method without action localization supervision. The reviewer also raised concerns, which we address below.
>
> **Q1:** Can the authors provide inference speed or computational cost analysis?
>
> **A1:** In addition to the dataset used in our main experiments, we conduct our experiments on SportsHHI , a recent benchmark focusing on human-human interaction in sports videos. We conducted an analysis of inference speed and computational cost on the SportsHHI dataset. Since the architectures used in prior works vary significantly, direct comparison is not straightforward. In our case, we integrated the disentanglement mechanism into a unified baseline architecture, which enables a fair comparison by isolating the performance improvements and overhead introduced specifically by the disentanglement module.
> We conduct two experiments for fair and thorough evaluation:
> - Module Integration: We build upon the official SportsHHI  baseline (implemented with MMAction2) by integrating our designed modules, including two Dynamic Prompt Modules (DPM)—one for the Unspecified Action Prompt (UAP) and one for the Specified Action Prompt (SAP)—as well as the proposed Action Disentanglement Loss (AD Loss).
> - Parameter-matched Baseline: To ensure a fair comparison, we expand the baseline model by increasing its parameters to match the model size of our method, without adding our modules.
>
> In the reported parameter count, we excluded the parametersof the Reconstruction Network, as it is not involved during inference. **For fair comparison, we reimplemented a version of the original model with larger parameter capacity, while keeping the architecture and training configuration consistent**
>
> As results shown in Table 1, our method introduces 10.8% more parameters than the baseline, but only increases inference time by 8.89%, indicating that the additional modules do not significantly compromise efficiency. More importantly, our method achieves a promising improvement in accuracy (+2.86 mAP). Furthermore, under our reimplemented version whose parameters are compatible with our method, our method still outperforms the baseline (+3.39 mAP),  demonstrating that the performance gain mainly comes from our proposed disentanglement mechanism rather than model size alone.
>
> SportsHHI  provides fine-grained annotations of interactions between players. SportsHHI  contains 34 high-level interaction classes from basketball and volleyball sports. 118,075 human bounding boxes and 50,649 interaction instances are annotated on 11,398 keyframes. Compared to conventional action datasets, SportsHHI offers richer relational labels and poses greater challenges due to dense, multi-person interactions
>
> **Table 1:**  Human-Human Interaction Classification results in SportsHHI.SportsHHI\* is our reimplementation based on the official SportsHHI code, with minor modifications (e.g., parameter size) to match the capacity of ProDA for fair comparison.
>
> |Method|mAP|R@150|R@100|R@50|R@20|Params|Time(s)|
> |-|-|-|-|-|-|-|-|
> |STTran| 3.31| 71.29|62.91|42.67|22.14|-|-|
> |HORT|3.75|78.57|67.78|50.33|26.96|-|-|
> |SlowFast|5.00| 81.66|74.00|52.74|26.82|-|-|
> |ACARN|5.44|82.85|75.22|56.53|31.77|-|-|
> |**SportsHHI**(baseline)|10.69|89.25|82.93|68.13|43.72|111.18M|0.0478|
> |**SportsHHI\***(reimplemented)|10.16 (↓0.53)| 90.07 (↑0.82) |84.83 (↑1.9)|68.40(↑0.37)|40.16(↓3.56)|123.11M|0.0484|
> |**Ours**(baseline+ProDA)|**13.55 (↑2.86)**|**92.54 (↑3.29)**|**87.91 (↑4.98)**|**72.69(↑4.56)**|**47.81(↑4.09)**|**123.19M**|**0.0520**|
>
> **Q2:** Could the authors include a qualitative visualization (e.g., t-SNE) comparing disentangled vs. unified features?
>
> **A2:** We appreciate the reviewer’s suggestion to include additional visualizations. However, due to the **rebuttal policy of NeurIPS 2025** which does not allow uploading additional figures, external links, or supplementary materials during the rebuttal stage, we are unable to provide the requested image at this time.
>
> We would be happy to include the full visualizations and corresponding analysis in the final version of the paper if it is accepted.
>
> **Although we are unable to show the visualization, we computed the inter-class and intra-class distances after applying t-SNE for dimensionality reduction**. For each video, we randomly disentangle one action, meaning each video has a unified feature and a disentangled feature, covering a total of 157 action classes. For the disentangled features, the intra-class and inter-class distances are **0.624 ± 0.438** and **46.769 ± 24.966**, respectively. For the unified features, we only computed the intra-class distance, which is **28.841**. It can be observed that the intra-class distances for the disentangled features are mostly within 1, while the inter-class distances are much larger. This indicates that our disentangled features effectively enlarge the distances between different action categories, making them more separable in the feature space. Meanwhile, the smaller intra-class distances suggest that features within the same category are more compact and consistent, enhancing intra-class stability. In contrast, the larger intra-class distance of the unified features indicates a more mixed representation with less distinct class boundaries. Overall, this validates the effectiveness of our feature disentanglement in separating actions, providing more discriminative representations for subsequent action recognition and understanding tasks.
>
>
> **Q3:** Why is the evaluation limited to a single dataset?
>
> **A3:** We thank the reviewer for pointing this out. In the original submission, we evaluated our method on Charades to focus on Multi-label classification. We have additionally evaluated our method on the SportsHHI dataset , which contains more diverse and complex human-human interaction scenarios on Human-Human-Interaction Classification (HHICls). In our original manuscript, we conducted experiments on Charades (Action Genome), which primarily focuses on human-object interactions. To further validate the generalizability of our method to more complex human-human interaction scenarios, we have added additional experiments on SportsHHI , a challenging benchmark designed specifically for dense and diverse human-human interactions in sports videos. Compared to Charades (Action Genome), SportsHHI involves more frequent and overlapping interactions among multiple players, better aligning with our target task and demonstrating the robustness of our approach across different interaction types.
>
> As results shown in **Table 1 above**. We observe that our method achieves a significant performance gain with only 10.80\% increase in parameters. Notably, our DPM modules contribute only 1.46M parameters, accounting for just 1.18\% of the total parameters. Under our reimplemented version with a parameter size comparable to our method, our approach still achieves a clear advantage (+3.39 mAP). This indicates that the performance improvement mainly stems from better action disentanglement rather than an increase in model capacity.
>
> **Q4:** How sensitive is ProDA to errors in SSG construction?
>
> **A4:** Thank you for the insightful question regarding the sensitivity of ProDA to noisy in the SSG construction. We follow the same setting as LaIAR , employing Faster R-CNN to detect and classify objects in videos, and to construct a noisy SSG (inaccurate object/relation predictions and missing temporal links for the same objects across frames ). As shown in Table 2, our method achieves the best performance even under the noisy SSG setting. Notably, the performance drop caused by the noisy SSG is smaller for our method compared to other approaches. We attribute this to our action disentanglement mechanism, which explicitly separates actions from the complex and error-prone scene context, allowing the model to reason over cleaner, disentangled representations. In contrast, feature extraction-based methods rely heavily on the error and entangled SSG, making them more sensitive to detection errors.
>
> **Table 2:** Results of SSGs under different conditions. GT SSGs are ground-truth SSGs without detection errors. FRCN SSGs are obtained using a detector (i.e., Faster R-CNN) and may contain errors.
> |Method|FRCNN SSGs mAP (%)|GT SSGs mAP (%)|
> |-|-|-|
> |LaIAR+NL|45.1|67.4 (+22.3)|
> |OR2G+NL|44.9|67.5 (+22.6)|
> |Ours (ProDA)|50.2| 71.1 (+20.9)|

---

> > ### Author Response · Authors · 2025-08-03
> >
> > Thank you again for your positive evaluation and thoughtful comments. We’ve carefully addressed your feedback in the rebuttal and would be grateful if you could let us know if there’s anything further we can clarify. We truly appreciate your support and the time you’ve dedicated to reviewing our work.

---

> > > ### Comment · Reviewer_MhYr · 2025-08-05
> > >
> > > Thank you to the authors for the thorough response. I commend the authors for their efforts in performing the additional experiments and presenting a candid and well-reasoned discussion that accurately reflects the new findings in light of the reviewers' suggestions.

---

> > > > ### Author Response · Authors · 2025-08-08
> > > >
> > > > Thank you again for your thoughtful and detailed review. We truly appreciate the time and effort you took to assess our work and provide valuable suggestions, which we have carefully incorporated in the revision.

---

### Note · Authors · 2025-08-12

Dear AC and Reviewers,

We thank you for your time and efforts during the review process. We are glad that the raised concerns have been addressed in our rebuttal, and we will incorporate the suggestions to further improve the final version.

Sincerely,
The Authors

---

### Decision · Program_Chairs · 2025-09-17

**Decision:**

Accept (poster)

**Comment:**

The paper proposes an approach that converts a short video clip into a scene graph. Given a pre-defined set of actions, it then identifies the sub-graph of the scene that corresponds to these actions. The reviewers appreciate that the problem, which is addressed in the paper, is well motivated and addresses a practical limitation of prior methods. Reviewer kb31 considers the idea of disentangling desired actions as compelling and acknowledges the thorough ablation studies. The reviewers, however, had several concerns. A major concern has been the experimental evaluation. It is performed on a single dataset and the impact of the quality of the spatio-temporal scene graph is not evaluated. There has been also concerns regarding the increased complexity compared to standard multi-label action recognition models and the unclear impact of the disentanglement loss, including a visualization that shows the disentanglement of the representation compared to conventional features. Finally, Reviewer SVW3 points out that the method does not use prompts but hot-vector encoded action  classes. The rebuttal provided additional results for the SportsHHI dataset and an analysis of the impact of the quality of the spatio-temporal scene graph. Preliminary results using text prompts have been provided as well, but these are not mature enough. After the rebuttal, the reviewers unanimously recommend the acceptance of the paper.

The AC acknowledges as the reviewers the additional results that have been reported in the rebuttal. Given the motivation of the problem and the shown benefit of the disentanglement loss, the AC agrees with the reviewers that the paper can be accepted. The manuscript, however, needs to be substantially improved by adding the new results, demonstrating the disentanglement (e.g. by adding a visualization), reconsidering the term 'prompt', and improving the reading flow as pointed out by Reviewer kb31.